# Neural Network Approximations of PDEs Beyond Linearity: Representational Perspective

## Abstract

A burgeoning line of research has developed deep neural networks capable of approximating the solutions to high dimensional PDEs, opening related lines of theoretical inquiry focused on explaining how it is that these models appear to evade the curse of dimensionality. However, most theoretical analyses thus far have been limited to simple linear PDEs. In this work, we take a step towards studying the representational power of neural networks for approximating solutions to nonlinear PDEs. We focus on a class of PDEs known as nonlinear variational elliptic PDEs, whose solutions minimize an Euler-Lagrange energy functional $\mathcal{E}(u) = \int_\Omega L(\nabla u) dx$. We show that if composing a function with Barron norm $b$ with $L$ produces a function of Barron norm at most $B_L b^p$, the solution to the PDE can be $\epsilon$-approximated in the $L^2$ sense by a function with Barron norm $O\left((dB_L)^{p^{\log(1/\epsilon)}}\right)$. By a classical result due to Barron (1993), this correspondingly bounds the size of a 2-layer neural network needed to approximate the solution. Treating $p, \epsilon, B_L$ as constants, this quantity is polynomial in dimension, thus showing that neural networks can evade the curse of dimensionality. Our proof technique involves "neurally simulating" (preconditioned) gradient in an appropriate Hilbert space, which converges exponentially fast to the solution of the PDE, and such that we can bound the increase of the Barron norm at each iterate. Our results subsume and substantially generalize analogous prior results for linear elliptic PDEs.

## 1 Introduction

Scientific applications have become one of the new frontiers for the application of deep learning (Jumper et al., 2021; Tunyasuvunakool et al., 2021; Sønderby et al., 2020). PDEs are one of the fundamental modeling techniques in scientific domains, and designing neural network-aided solvers, particularly in high-dimensions, is of widespread usage in many domains (Hsieh et al., 2019; Brandstetter et al., 2022). One of the most common approaches for applying neural networks to solve PDEs is to parameterize the solution as a neural network and minimize a loss which characterizes the solution (Sirignano & Spiliopoulos, 2018; E & Yu, 2017). The hope in doing so is to have a method which computationally avoids the "curse of dimensionality"—i.e., that scales less than exponentially with the ambient dimension.

To date, neither theoretical analysis nor empirical applications have yielded a precise characterization of the range of PDEs for which neural network-aided methods outperform classical methods. Active research on the empirical side (Han et al., 2018; E et al., 2017; Li et al., 2020a;b) has explored several families of PDEs, e.g., Hamilton-Bellman-Jacobi and Black-Scholes, where neural networks have been demonstrated to outperform classical grid-based methods. On the theory side, a recent line of works (Marwah et al., 2021; Chen et al., 2021; 2022) has considered the following fundamental question:

*For what families of PDEs can the solution be represented by a small neural network?*

The motivation for this question is computational: since the computational complexity of fitting a neural network (by minimizing some objective) will grow with its size. Specifically, these works focus on understanding when the approximating neural network can be sub-exponential in size, thus

avoiding the curse of dimensionality. Unfortunately, the techniques introduced in this line of work have so far only been applicable to *linear* PDEs.

In this paper, we take the first step beyond linear PDEs, with a particular focus on *nonlinear variational elliptic PDEs*. These equations have the form $-\text{div}(\nabla L(\nabla u)) = 0$ and are instances of *nonlinear Euler-Lagrange* equations. Equivalently, $u$ is the minimizer of the energy functional $\mathcal{E}(u) = \int_\Omega L(\nabla u)dx$. This paradigm is very generic: its origins are in Lagrangian formulations of classical mechanics, and for different $L$, a variety of variational problems can be modeled or learned (Schmidt & Lipson, 2009; Cranmer et al., 2020). These PDEs have a variety of applications in scientific domains, e.g., (non-Newtonian) fluid dynamics (Koleva & Vulkov, 2018), meteorology(Weller et al., 2016), and nonlinear diffusion equations (Burgers, 2013).

Our main result is to show that *when the function L has "low complexity", so does the solution*. The notion of complexity we work with is the *Barron norm* of the function, similar to Chen et al. (2021); Lee et al. (2017). This is a frequently used notion of complexity, as a function with small Barron norm can be represented by a small, two-layer neural network, due to a classical result (Barron, 1993). Mathematically, our proof techniques are based on "neurally unfolding" an iterative preconditioned gradient descent in an appropriate function space: namely, we show that each of the iterates can be represented by a neural network with Barron norm not much worse than the Barron norm of the previous iterate—along with showing a bound on the number of required steps.

Importantly, our results go beyond the typical non-parametric bounds on the size of an approximator network that can be easily shown by classical regularity results of the solution to the nonlinear variational PDEs (De Giorgi, 1957; Nash, 1957; 1958) along with universal approximation results (Yarotsky, 2017).

## 2 OVERVIEW OF RESULTS

Let $\Omega \subset \mathbb{R}^d$ be a bounded open set with $0 \in \Omega$ and $\partial\Omega$ denote the boundary of $\Omega$. Furthermore, we assume that the domain $\Omega$ is such that the Poincare constant $C_p$ is greater than 1 (see Theorem 2 for the exact definition of the Poincare constant).

We first define the energy functional whose minimizers are represented by a nonlinear variational elliptic PDE—i.e., the Euler-Lagrange equation of the energy functional.

**Definition 1** (Energy functional). *For all $u: \Omega \to \mathbb{R}$ such that $u|_{\partial\Omega} = 0$, we consider an energy functional of the following form:*

$$\mathcal{E}(u) = \int_\Omega L(\nabla u)dx, \tag{1}$$

*where $L : \mathbb{R}^d \to \mathbb{R}$ is a smooth and uniformly convex function , i.e., there exists constant $0 < \lambda \leq \Lambda$ such that for all $x \in \mathbb{R}$ we have $\lambda I_d \leq D^2 L(x) \leq \Lambda I_d$. Further, without loss of generality[1], we assume that $\lambda \leq 1/C_p$.*

Note that due to the convexity of the function $L$, the minimizer $u^\star$ exists and is unique. The proof of existence and uniqueness is standard (e.g., Theorem 3.3 in Fernández-Real & Ros-Oton (2020)).

Writing down the condition for stationarity, we can derive a (nonlinear) elliptic PDE for the minimizer of the energy functional in Definition 1 .

**Lemma 1.** *Let $u^\star : \Omega \to \mathbb{R}$ be the unique minimizer for the energy functional in Definition 1. Then for all $\varphi \in H_0^1(\Omega)$ the minimizer $u^\star$ satisfies the following condition,*

$$D\mathcal{E}[u](\varphi) = \int_\Omega \nabla L(\nabla u)\nabla\varphi dx = 0, \tag{2}$$

*where $d\mathcal{E}[u](\varphi)$ denotes the dirctional derivative of the energy functional calculated at $u$ in the direction of $\varphi$. Thus, the minimizer $u^\star$ of the energy functional satisfies the following PDE:*

$$D\mathcal{E}(u) := -\text{div}(\nabla L(\nabla u)) = 0 \qquad \forall x \in \Omega. \tag{3}$$

*and $u(x) = 0, \forall x \in \partial\Omega$. Here $\text{div}$ denote the divergence operator.*

---

[1]Since $\lambda$ is a lower bound on the strong convexity constant. If we choose a weaker lower bound, we can always ensure $\lambda \leq 1/C_p$.

The proof for the Lemma can be found in Appendix Section A.1. Here, $-\mathrm{div}(\nabla L(\nabla \cdot)$ is a functional operator that acts on a function (in this case $u$). [2]

Our goal is to determine if the solution to the PDE in Equation 3 can be expressed by a neural network with a small number of parameters. In order do so, we utilize the concept of a *Barron norm*, which measures the complexity of a function in terms of its Fourier representation. We show that if composing with the function $L$ is such that it increases it has a bounded increase in the Barron norm of $u$, then the solution to the PDE in Equation 3 will have a bounded Barron norm. The motivation for using this norm is a seminal paper (Barron, 1993), which established that any function with Barron norm $C$ can be $\epsilon$-approximated in the $L^2$ sense by a two-layer neural network with size $O(C^2/\epsilon)$, thus evading the curse of dimensionality if $C$ is substantially smaller than exponential in $d$. Informally, we will show the following result:

**Theorem 1** (Informal). *Let $L$ be convex and smooth, such that composing a function with Barron norm $b$ with $L$ produces a function of Barron norm at most $B_L b^p$. Then, for all sufficiently small $\epsilon > 0$, the minimizer of the energy functional in Definition 1 can be $\epsilon$-approximated in the $L^2$ sense by a function with Barron norm $O\left((dB_L)^{p^{\log(1/\epsilon)}}\right)$.*

As a consequence, when $\epsilon, p$ are thought of as constants, we can represent the solution to the Euler-Lagrange PDE Equation 3 by a polynomially-sized network, as opposed to an exponentially sized network, which is what we would get by standard universal approximation results and using regularity results for the solutions of the PDE.

We establish this by "neurally simulating" a preconditioned gradient descent (for a strongly-convex loss) in an appropriate Hilbert space, and show that the Barron norm of each iterate—which is a function—is finite, and at most polynomially bigger than the Barron norm of the previous iterate. We get the final bound by (i) bounding the growth of the Barron norm at every iteration; and (ii) bounding the number of iterations required to reach an $\epsilon$-approximation to the solution. The result in formally stated in Section 5

## 3 RELATED WORK

Over the past few years, a growing line of work has focused on parameterizing the solutions to PDEs with neural networks. Works such as E et al. (2017); E & Yu (2017); Sirignano & Spiliopoulos (2018); Raissi et al. (2017) achieved impressive results on a variety of different applications and demonstrated the empirical efficacy of neural networks in solving high-dimensional PDEs, even outperforming previously dominant numerical approaches like finite differences and finite element methods (LeVeque, 2007) that proceed by discretizing the input space, hence limiting their use to problems on low-dimensional input spaces.

Several recent works look to theoretically analyze these neural network based approaches for solving PDEs. Mishra & Molinaro (2020) look at the generalization properties of physics informed neural networks. In Lu et al. (2021) show the generalization analysis for the Deep Ritz method for elliptic equations like the Poisson equation and Lu & Lu (2021) extends their analysis to the Schrödinger eigenvalue problem.

In addition to analyzing the generalization capabilities of the neural networks, theoretical analysis into their representational capabilities has also gained a lot of attention. Khoo et al. (2021) show the existence of a network by discretizing the input space into a mesh and then using convolutional NNs, where the size of the layers is exponential in the input dimension. Sirignano & Spiliopoulos (2018) provide a universal approximation result, showing that for sufficiently regularized PDEs, there exists a multilayer network that approximates its solution. In Jentzen et al. (2018); Grohs & Herrmann (2020); Hutzenthaler et al. (2020) provided a better-than-exponential dependence on the input dimension for some special parabolic PDEs, based on a stochastic representation using the Feynman-Kac Lemma, thus limiting the applicability of their approach to PDEs that have such a probabilistic interpretation. Moreover, their results avoid the curse of dimensionality only over domains with unit volume.

---

[2]For a vector valued function $F : \mathbb{R}^d \to \mathbb{R}^d$, we will denote the divergence operator either by $\mathrm{div}F$ or by $\nabla \cdot F$, where $\mathrm{div}F = \nabla \cdot F = \sum_{i=1}^{d} \frac{\partial_i F}{\partial x_i}$.

A recent line of work has focused on families of PDEs for which neural networks evade the curse of dimensionality—i.e., the solution can be approximated by a neural network with a subexponential size. Marwah et al. (2021) show that for elliptic PDE's whose coefficients are approximable by neural networks with at most $N$ parameters, a neural network exists that $\epsilon$-approximates the solution and has size $O(d^{\log(1/\epsilon)}N)$. Chen et al. (2021) extends this analysis to elliptic PDEs with coefficients with small Barron norm, and shows that if the coefficients have Barron norm bounded by $B$, an $\epsilon$-approximate solution exists with Barron norm at most $O(d^{\log(1/\epsilon)}B)$. The work by Chen et al. (2022) derives related results for the Schrödinger equation.

As mentioned, while most previous works show key regularity results for neural network approximations of solution to PDEs, most of their analysis is limited to simple *linear* PDEs. The focus of this paper is towards extending these results to a family of PDEs referred to as nonlinear variational PDEs. This particular family of PDEs consists of many famous PDEs, such as $p-$Laplacian (on a bounded domain), which is used to model phenomena like non-Newtonian fluid dynamics and non-linear diffusion processes. The regularity results for these family of PDEs was posed as Hilbert's XIX$^{th}$ problem. We note that there are classical results like De Giorgi (1957) and Nash (1957; 1958) that provide regularity estimates on the solutions of a nonlinear variational elliptic PDE of the form in Equation 3. One can easily use these regularity estimates, along with standard universal approximation results (Yarotsky, 2017) to show that the solutions can be approximated arbitrarily well. However, the size of the resulting networks will be exponentially large (i.e. suffer from the curse of dimensionality)—so are of no use for our desired results.

## 4 NOTATION AND DEFINITION

In this section, we introduce some key concepts and notation that will be used throughout the paper. For a vector $x \in \mathbb{R}^d$, we use $\|x\|_2$ to denote its $\ell_2$ norm. Further, $C^\infty(\Omega)$ denotes the set of functions $f : \Omega \to \mathbb{R}$ that are infinitely differentiable. We also define some important function spaces and associated key results below.

**Definition 2.** *For a vector valued function $g \colon \mathbb{R} \to \mathbb{R}^d$, we define the $L^p(\Omega)$ norm for $p \in [1, \infty)$ as*

$$\|g\|_{L^p(\Omega)} = \left( \int_\Omega \sum_i^d |g_i(x)|^p \, dx \right)^{1/p},$$

*For $p = \infty$, we have*

$$\|g\|_{L^\infty(\Omega)} = \max_{1 \leq i \leq d} \|g_i\|_{L^\infty(\Omega)},$$

*where $\|g_i\|_{L^\infty(\Omega)} = \inf\{c \geq 0 : |g(x)| \leq c \text{ for almost all } x \in \Omega\}$.*

**Definition 3.** *For a domain $\Omega$, the space of functions $H_0^1(\Omega)$ is defined as*

$$H_0^1(\Omega) := \{g : \Omega \to \mathbb{R} : g \in L^2(\Omega), \nabla g \in L^2(\Omega), g|_{\partial\Omega} = 0\}.$$

*The corresponding norm for $H_0^1(\Omega)$ is defined as $\|g\|_{H_0^1(\Omega)} = \|\nabla g\|_{L^2(\Omega)}$.*

We will make use of the Poincaré inequality throughout several of our results.

**Theorem 2** (Poincaré inequality, Poincaré (1890)). *For $\Omega \subset \mathbb{R}^d$ which is open and bounded, there exists a constant $C_p > 0$ such that for all $u \in H_0^1(\Omega)$*

$$\|u\|_{L^2(\Omega)} \leq C_p \|\nabla u\|_{L^2(\Omega)}.$$

This constant can be very benignly behaved with dimension for many natural domains—even dimension independent. Examples include convex domains (Payne & Weinberger, 1960).

### 4.1 BARRON NORMS

For a function $f : \mathbb{R}^d \to \mathbb{R}$, the Fourier transform and the inverse Fourier transform are defined as

$$\hat{f}(\omega) = \frac{1}{(2\pi)^d} \int_{\mathbb{R}^d} f(x)e^{-ix^T\omega}dx, \quad \text{and} \quad f(x) = \int_{\mathbb{R}^d} \hat{f}(\omega)e^{ix^T\omega}d\omega. \tag{4}$$

The Barron norm is an average of the norm of the frequency vector weighted by the Fourier magnitude $|\hat{f}(\omega)|$. A slight technical issue is that the the Fourier transform is defined only for $f : \mathbb{R}^d \to \mathbb{R}$. Since we are interested in defining the Barron norm of functions defined over a bounded domain, we allow for arbitrary extensions of a function outside of their domain. (This is the standard definition, e.g. in (Barron, 1993).)

**Definition 4.** *We define $\mathcal{F}$ be the set of functions $g \in L^1(\Omega)$ such that the Fourier inversion formula $g = (2\pi)^d \hat{f}(x)$ holds over the domain $\Omega$, i.e.,*

$$\mathcal{F} = \left\{ g : \mathbb{R}^d \to \mathbb{R}, \forall x \in \Omega, g(x) = g(0) + \int_{\mathbb{R}^d} (e^{i\omega^T x} - 1)\hat{g}(\omega)d\omega \right\}.$$

**Definition 5** (Spectral Barron Norm, (Barron, 1993)). *Let $\Gamma$ be a set of functions defined over $\Omega$ such that their extension over $\mathbb{R}^d$ belong to $\mathcal{F}$, that is,*

$$\Gamma = \{ f : \Omega \to \mathbb{R} : \exists g, g|_\Omega = f, g \in \mathcal{F} \}$$

*Then we define the spectral Barron norm $\| \cdot \|_{\mathcal{B}(\Omega)}$ as*

$$\|f\|_{\mathcal{B}(\Omega)} = \inf_{g|_\Omega = f, g \in \mathcal{F}} \int_{\mathbb{R}^d} (1 + \|\omega\|_2)|\hat{g}(\omega)|d\omega.$$

The Barron norm is an $L_1$ relaxation of requiring sparsity in the Fourier basis—which is intuitively why it confers representational benefits in terms of the size of a neural network required. We refer to Barron (1993) for a more exhaustive list of the Barron norms of some common function classes.

The main theorem from Barron (1993) formalizes this intuition, by bounding the size of a 2-layer network approximating a function with small Barron norm:

**Theorem 3** (Theorem 1, Barron (1993)). *Let $f \in \Gamma$ such that $\|f\|_{\mathcal{B}(\Omega)} \leq C$ and $\mu$ be a probability measure defined over $\Omega$. There exists $a_i \in \mathbb{R}^d$, $b_i \in \mathbb{R}$ and $c_i \in \mathbb{R}$ such that $\sum_{i=1}^k |c_i| \leq 2C$, there exists a function $f_k(x) = \sum_{i=1}^k c_i \sigma\left(a_i^T x + b\right)$, such that we have,*

$$\int_\Omega (f(x) - f_k(x))^2 \mu(dx) \leq \frac{4C^2}{k}.$$

*Here $\sigma$ denotes a sigmoidal activation function, i.e., $\lim_{x \to \infty} \sigma(x) = 1$ and $\lim_{x \to -\infty} \sigma(x) = 0$.*

Note that while Theorem 3 is stated for sigmoidal activations like *sigmoid* and *tanh* (after appropriate rescaling), the results are also valid for ReLU activation functions, since $\text{ReLU}(x) - \text{ReLU}(x - 1)$ is in fact sigmoidal. We will also need to work with functions that do not have Fourier coefficients beyond some size (i.e. are band limited), hence we introduce the following definition:

**Definition 6.** *Let $\mathcal{F}_W(\Omega)$ be the set of functions whose Fourier coefficients vanish outside a bounded ball, i.e.,*

$$\mathcal{F}_W = \{ g : \mathbb{R}^d \to \mathbb{R} : s.t. \ \forall w, \|w\|_\infty \geq W, \hat{g}(w) = 0 \}.$$

*Similarly, we denote*

$$\Gamma_W = \left\{ f : \Omega \to \mathbb{R} : \exists g, g|_\Omega = f, g \in \mathcal{F}_W \right\}.$$

Since we will work with vector valued function, we will also define the Barron norm of a vector-valued function as the maximum of the Barron norms of its coordinates:

**Definition 7.** *For a vector valued function $g : \Omega \to \mathbb{R}^d$, we define $\|g\|_{\mathcal{B}(\Omega)} = \max_i \|g_i\|_{\mathcal{B}(\Omega)}$.*

## 5 MAIN RESULT

Before stating the main result we introduce the key assumption.

**Assumption 1.** *The function $L$ in Definition 1 can be approximated by a function $\tilde{L} : \mathbb{R}^d \to \mathbb{R}$ such that there exists a constant $\epsilon_L \in [0, \lambda)$ with $\sup_{x \in \mathbb{R}^d} \|\nabla L(x) - \nabla \tilde{L}(x)\|_2 \leq \epsilon_L \|x\|_2$.*

*Furthermore, we assume that $\tilde{L}$ is such that for any $g \in H_0^1(\Omega)$, we have $\tilde{L} \circ g \in H_0^1(\Omega)$, $\tilde{L} \circ g \in \mathcal{F}$ and*

$$\|\tilde{L} \circ g\|_{\mathcal{B}(\Omega)} \leq B_{\tilde{L}} \|g\|_{\mathcal{B}(\Omega)}^p. \tag{5}$$

*for some constants $B_{\tilde{L}} \geq 0$, and $p \geq 0$. Furthermore, if $g \in \Gamma_W$ then $\tilde{L} \circ g \in \Gamma_{kW}$ for a $k > 0$.*

This assumption is fairly natural: it states that the function $L$ can be approximated (up to $\epsilon_L$, in the sense of the gradients of the functions) by a function $\tilde{L}$ that has the property that when applied to a function $g$ with small Barron norm, the new Barron norm is not much bigger. The constant $p$ specifies the order of this growth. The functions for which our results are most interesting are when the dependence of $B_{\tilde{L}}$ on $d$ is at most polynomial—so that the final size of the approximating network does not exhibit curse of dimensionality. For instance, we can take $L$ to be a multivariate polynomial of degree up to $P$: we show in Lemma 7 the constant $B_{\tilde{L}}$ is $O(d^P)$ (intuitively, this dependence comes from the total number of monomials of this degree), whereas $p$ and $k$ are both $O(P)$.

With all the assumptions stated, we now state our main theorem:

**Theorem 4** (Main Result). *Consider the nonlinear variational elliptic PDE in Equation 3 which satisfies Assumption 1 and let $u^\star \in H_0^1(\Omega)$ denote the solution to the PDE. If $u_0 \in H_0^1(\Omega)$ is a function such that $u_0 \in \Gamma_{W_0}$, then for all sufficiently small $\epsilon > 0$, and*

$$T := \left\lceil \log \left( \frac{2}{\epsilon} \frac{(\mathcal{E}(u_0) - \mathcal{E}(u^\star))}{\lambda} \right) \Big/ \log \left( \frac{1}{1 - \frac{\lambda^5}{(1+C_p)\Lambda^4}} \right) \right\rceil,$$

*there exists a function $u_T \in H_0^1(\Omega)$ such that $u_T \in \Gamma_{k^T W_0}$ and Barron norm bounded as*

$$\|u_T\|_{\mathcal{B}(\Omega)} \leq \left( 1 + \frac{\lambda^3}{(C_p+1)\Lambda^3} dk^2 W_0^2 B_{\tilde{L}} \right)^{p^T+1} \|u_0\|_{\mathcal{B}(\Omega)}^{p^T}.$$

*Furthermore, $u_T$ satisfies $\|u_T - u^\star\|_{H_0^1(\Omega)} \leq \epsilon + \tilde{\epsilon}$ where*

$$\tilde{\epsilon} \leq \frac{\lambda^3}{(C_p+1)\Lambda^3} \frac{\epsilon_L \left( \|u^\star\|_{H_0^1(\Omega)} + \frac{1}{\lambda} \mathcal{E}(u_0) \right)}{\Lambda + \epsilon_L} \left( \left( 1 + \frac{\lambda^3}{(C_p+1)\Lambda^3} (\Lambda + \epsilon_L) \right)^T - 1 \right).$$

*Remark 1:* The function $u_0$ can be seen as an initial estimate of the solution, that can be refined to an estimate $u_T$, which is progressively better at the expense of a larger Barron norm. A trivial choice could be $u_0 = 0$, which has Barron norm 1, and which by Lemma 2 would satisfy $\mathcal{E}(u_0) - \mathcal{E}(u^*) \leq \Lambda \|u^*\|_{H_0^1(\Omega)}^2$.

*Remark 2:* The final approximation error has two terms, $T$ goes to $\infty$ as $\epsilon$ tends 0 and is a consequence of the way $u_T$ is constructed—by simulating a functional (preconditioned) gradient descent which converges to the solution to the PDE. The error term $\tilde{\epsilon}$ stems from the approximation that we make between $\tilde{L}$ and $L$, which grows as $T$ increases—it is a consequence of the fact that the gradient descent updates with $\tilde{L}$ and $L$ progressively drift apart as $T$ tends to $\infty$.

*Remark 3:* As in the informal theorem, if we think of $p, \Lambda, \lambda, C_p, k, \|u_0\|_{\mathcal{B}(\Omega)}$ as constants, the theorem implies $u^*$ can be $\epsilon$-approximated in the $L^2$ sense by a function with Barron norm $O\left( (dB_L)^{p^{\log(1/\epsilon)}} \right)$. Therefore, combining results from Theorem 4 and Theorem 3 the total parameters required to $\epsilon$-approximate $u^*$ by a 2-layer neural network is $O\left( \frac{1}{\epsilon^2} (dB_L)^{2p^{\log(1/\epsilon)}} \right)$.

*Remark 4:* We further note that this result recovers (and generalizes) prior results which bound the Barron norm of linear elliptic PDEs like Chen et al. (2021). In these results, the elliptic PDE takes the form $-\text{div}(A\nabla u)$ and $A$ is assumed to have bounded Barron norm. Thus, $\|L \circ u\|_{\mathcal{B}(\Omega)} \leq d^2 \|A\|_{\mathcal{B}(\Omega)} \|u\|_{\mathcal{B}(\Omega)}$, hence satisfying Equation 5 in Assumption 1 with $p = 1$.

## 6 PROOF OF MAIN RESULT

The proof will proceed by "neurally unfolding" a preconditioned gradient descent on the objective $\mathcal{E}$ in the Hilbert space $H_0^1(\Omega)$. This is inspired by previous works by Marwah et al. (2021); Chen et al. (2021) where the authors show that for a linear elliptic PDE, an objective which is quadratic can be designed. In our case, we show that $\mathcal{E}$ is "strongly convex" in some suitable sense—thus again, bounding the amount of steps needed.

More precisely, the result will proceed in two parts:

1. First, we will show that the sequence of functions $\{u_t\}_{t=0}^{\infty}$, where $u_{t+1} \leftarrow u_t - \eta(I - \Delta)^{-1} d\mathcal{E}(u_t)$ can be interpreted as performing preconditioned gradient descent, with the (constant) preconditioner $(I - \Delta)^{-1}$. We show that in some appropriate sense (Lemma 2), $\mathcal{E}$ is strongly convex in $H_0^1(\Omega)$—thus the updates converge at a rate of $O(\log(1/\epsilon))$.

2. We then show that the Barron norm of each iterate $u_{t+1}$ can be bounded in terms of the Barron norm of the prior iterate $u_t$. We show this in Lemma 5, where we show that given Assumptions1, the $\|u_{t+1}\|_{\mathcal{B}(\Omega)}$ is $O(d\|u_t\|_{\mathcal{B}(\Omega)}^p)$. By unrolling this recursion we show that the Barron norm of the $\epsilon$-approximation of $u^\star$ is of the order $O(d^{p^T}\|u_0\|_{\mathcal{B}(\Omega)}^p)$, where $T$ are the total steps required for $\epsilon$-approximation and $\|u_0\|_{\mathcal{B}(\Omega)}$ is the Barron norm of the first function in the iterative updates.

We now proceed to delineate the main technical ingredients for both of these parts.

## 6.1 CONVERGENCE RATE OF SEQUENCE

The proof to show the convergence to the solution $u^\star$ is based on adapting the standard proof (in finite dimension) for convergence of gradient descent when minimizing a strongly convex function $f$. Recall, the basic idea is to Taylor expand $f(x + \delta) \approx f(x) + \nabla f(x)^T \delta + O(\|\delta\|^2)$. Taking $\delta = \eta \nabla f(x)$, we lower bound the progress term $\eta \|\nabla f(x)\|^2$ using the convexity of $f$, and upper bound the second-order term $\eta^2 \|\nabla f(x)\|^2$ using the smoothness of $f$.

We follow analogous steps, and prove that we can lower bound the progress term by using some appropriate sense of convexity of $\mathcal{E}$, and upper bound some appropriate sense of smoothness of $\mathcal{E}$, when considered as a function over $H_0^1(\Omega)$. Precisely, we show:

**Lemma 2** (Strong convexity of $\mathcal{E}$ in $H_0^1$). *If $\mathcal{E}, L$ are as in Definition 1, we have*

1. $\forall u, v \in H_0^1(\Omega) : \langle D\mathcal{E}(u), v \rangle_{L^2(\Omega)} = \int_\Omega -\mathrm{div}(\nabla L(\nabla u))v dx = \int_\Omega \nabla L(\nabla u) \cdot \nabla v dx.$

2. $\forall u, v \in H_0^1(\Omega) : \lambda \|u - v\|_{H_0^1(\Omega)} \le \langle D\mathcal{E}(u) - D\mathcal{E}(v), u - v \rangle_{L^2(\Omega)} \le \Lambda \|u - v\|_{H_0^1(\Omega)}.$

3. $\forall u, v \in H_0^1(\Omega) : \frac{\lambda}{2}\|\nabla v\|_{L^2(\Omega)}^2 + \langle D\mathcal{E}(u), v \rangle_{L^2(\Omega)} \le \mathcal{E}(u+v) - \mathcal{E}(u) \le \langle D\mathcal{E}(u), v \rangle_{L^2(\Omega)} + \frac{\Lambda}{2}\|\nabla v\|_{L^2(\Omega)}^2.$

4. $\forall u \in H_0^1(\Omega) : \frac{\lambda}{2}\|u - u^\star\|_{H_0^1(\Omega)}^2 \le \mathcal{E}(u) - \mathcal{E}(u^\star) \le \frac{\Lambda}{2}\|u - u^\star\|_{H_0^1(\Omega)}^2.$

Part 1 is a helpful way to rewrite an inner product of a "direction" $v$ with $D\mathcal{E}(u)$—it is essentially a consequence of integration by parts and the Dirichlet boundary condition. Part 2 and 3 are common proxies of convexity: they are ways of formalizing the notion that $\mathcal{E}$ is strongly convex, when viewed as a function over $H_0^1(\Omega)$. Finally, part 4 is a consequence of strong convexity, capturing the fact that if the value of $\mathcal{E}(u)$ is suboptimal, $u$ must be (quantitatively) far from $u^*$. The proof of the Lemma can be found in the Appendix (Section B.1)

When analyzing gradient descent (in finite dimensions) to minimize a loss function $\mathcal{E}$, the standard condition for progress is that the inner product of the gradient with the direction towards the optimum is lower bounded as $\langle D\mathcal{E}(u), u^* - u \rangle_{L^2(\Omega)} \ge \alpha \|u - u^*\|_{L^2(\Omega)}^2$. From Parts 2 and 3, one can readily see that the above condition is only satisfied "with the wrong norm": i.e. we only have $\langle D\mathcal{E}(u), u^* - u \rangle_{L^2(\Omega)} \ge \alpha \|u - u^*\|_{H_0^1(\Omega)}^2$. We can fix this mismatch by instead doing preconditioned gradient, using the fixed preconditioner $(I - \Delta)^{-1}$. Towards that, the main lemma about the preconditioning we require is the following one:

**Lemma 3** (Norms with preconditioning). *For all $v \in H_0^1(\Omega)$, we have*

1. $\|(I - \Delta)^{-1}\nabla \cdot \nabla u\|_{L^2(\Omega)} = \|(I - \Delta)^{-1}\Delta u\|_{L^2(\Omega)} \le \|u\|_{L^2(\Omega)}.$

2. $\langle (I - \Delta)^{-1} v, v \rangle_{L^2(\Omega)} \ge \frac{1}{1+C_p} \langle \Delta^{-1} v, v \rangle_{L^2(\Omega)}.$

The first part of the lemma is a relatively simple consequence of the fact that the operators $\Delta$ and $\nabla$ "commute", and therefore can be re-ordered. The latter lemma can be understood intuitively as

$(I - \Delta)^{-1}$ and $\Delta^{-1}$ act as similar operators on eigenfunctions of $\Delta$ with large eigenvalues (the extra $I$ does not do much)—and are only different for eigenfunctions for small eigenvalues. However, since the smallest eigenvalues is lower bounded by $1/C_p$, their gap can be bounded.

Next we utilize the results in Lemma 2 and Lemma 3 to show preconditioned gradient descent exponentially converges to the solution to the nonlinear variational elliptic PDE in 3.

**Lemma 4** (Preconditioned Gradient Descent Convergence). *Let $u^\star$ denote the unique solution to the PDE in Definition 3. For all $t \in \mathbb{N}$, we define the sequence of functions*

$$u_{t+1} \leftarrow u_t - \frac{\lambda^3}{(1 + C_p)\Lambda^3}(I - \Delta)^{-1}D\mathcal{E}(u_t). \tag{6}$$

*If $u_0 \in H_0^1(\Omega)$ after $t$ iterations we have,*

$$\mathcal{E}(u_{t+1}) - \mathcal{E}(u^\star) \leq \left(1 - \frac{\lambda^5}{(1 + C_p)\Lambda^4}\right)^t (\mathcal{E}(u_0) - \mathcal{E}(u^\star)).$$

The complete proof for convergence can be found in Section B.3 of the Appendix.

Therefore, using the result from Lemma 4, i.e., $\|u_t - u^\star\|_{H_0^1(\Omega)}^2 \leq \frac{2}{\lambda}(\mathcal{E}(u_t) - \mathcal{E}(u^\star))$, we have

$$\|u_t - u^\star\|_{H_0^1(\Omega)}^2 \leq \frac{2}{\lambda}\left(1 - \frac{\lambda^5}{2(1 + C_p)\Lambda^4}\right)^t (\mathcal{E}(u_0) - \mathcal{E}(u^\star)).$$

and $\|u_T - u^\star\|_{H_0^1(\Omega)}^2 \leq \epsilon$ after $T$ steps, where,

$$T \geq \log\left(\frac{\mathcal{E}(u_0) - \mathcal{E}(u^\star)}{\lambda\epsilon/2}\right) / \log\left(\frac{1}{1 - \frac{\lambda^5}{(1+C_p)\Lambda^4}}\right). \tag{7}$$

## 6.2 BOUNDING THE BARRON NORM

Having obtained a sequence of functions that converge to the solution $u^\star$, we bound the Barron norms of the iterates. We draw inspiration from Marwah et al. (2021); Lu et al. (2021) and show that the Barron norm of each iterate in the sequence has a bounded increase on the Barron norm of the previous iterate. Note that in general, the Fourier spectrum of a composition of functions can not easily be expressed in terms of the Fourier spectrum of the functions being composed. However, from Assumption 1, we know that the function $L$ can be approximated by $\tilde{L}$ such that $\tilde{L} \circ g$ has a bounded increase the Barron norm of $g$. Thus, instead of tracking the iterates in Equation 6, we track the Barron norm of the functions in the following sequence,

$$\tilde{u}_{t+1} = \tilde{u}_t - \eta(I - \Delta)^{-1}D\tilde{\mathcal{E}}(\tilde{u}_t). \tag{8}$$

We can derive the following result (the proof is deferred to Section D.1 of the Appendix):

**Lemma 5.** *Consider the updates in Equation 8, if $\tilde{u}_t \in \Gamma_{W_t}$ then for all $\eta \in (0, \frac{\lambda^3}{(C_p+1)\Lambda^3}]$ we have $\tilde{u}_{t+1} \in \Gamma_{kW_t}$ and the Barron norm of $\tilde{u}_{t+1}$ can be bounded as*

$$\|\tilde{u}_{t+1}\|_{\mathcal{B}(\Omega)} \leq \left(1 + \eta d(kW_t)^2 B_{\tilde{L}}\right)\|\tilde{u}_t\|_{\mathcal{B}(\Omega)}^p.$$

The proof consists of using the result in Equation 5 about the Barron norm of composition of a function with $\tilde{L}$, as well as counting the increase in the Barron norm of a function by any basic algebraic operation, as established in Lemma 6. Precisely we show:

**Lemma 6** (Barron norm algebra). *If $h, h_1, h_2 \in \Gamma$, then the following set of results hold,*

- *Addition: $\|h_1 + h_2\|_{\mathcal{B}(\Omega)} \leq \|h_1\|_{\mathcal{B}(\Omega)} + \|h_2\|_{\mathcal{B}(\Omega)}$ .*

- *Multiplication: $\|h_1 \cdot h_2\|_{\mathcal{B}(\Omega)} \leq \|h_1\|_{\mathcal{B}(\Omega)}\|h_2\|_{\mathcal{B}(\Omega)}$*

- *Derivative: if $h \in \Gamma_W$ for $i \in [d]$ we have $\|\partial_i h\|_{\mathcal{B}(\Omega)} \in W\|h\|_{\mathcal{B}(\Omega)}$.*

- *Preconditioning: if $h \in \Gamma$, then $\|(I - \Delta)^{-1}h\|_{\mathcal{B}(\Omega)} \le \|h\|_{\mathcal{B}(\Omega)}$.*

The proof for the above Lemma can be found in Appendix D.3. It bears similarity to an analogous result in Chen et al. (2021), with the difference being that our bounds are defined in the *spectral* Barron space which is different from the definition of the Barron norm used in Chen et al. (2021).

Expanding on the recurrence in Lemma 6 we therefore have that after $T$ we have, we have $W_T \le k^T W_0$ and hence $u_{t+1} \in \Gamma_{k^t W_0}$, and the Barron norm of $u_T$ can be bounded as follows,

$$\|u_T\|_{\mathcal{B}(\Omega)} \le \left(1 + \eta d k^2 W_0^2 B_{\tilde{L}}\right)^{p^T+1} \|u_0\|_{\mathcal{B}(\Omega)}^{p^T}. \tag{9}$$

Finally, we exhibit a natural class of functions that satisfy the main Barron growth property in Equations 5. Precisely, we show (multivariate) polynomials of bounded degree have an effective bound on $p$ and $B_L$:

**Lemma 7.** *Let $f(x) = \sum_{\alpha,|\alpha| \le P} \left(A_\alpha \prod_{i=1}^d x_i^{\alpha_i}\right)$ where $\alpha$ is a multi-index and $x \in \mathbb{R}^d$. If $g : \mathbb{R}^d \to \mathbb{R}^d$ is such that $g \in \Gamma_W$, then we have $f \circ g \in \Gamma_{PW}$ and the Barron norm can be bounded as, $\|f \circ g\|_{\mathcal{B}(\Omega)} \le d^{P/2} \left(\sum_{\alpha,|\alpha| \le P} |A_\alpha|^2\right)^{1/2} \|g\|_{\mathcal{B}(\Omega)}^P$.*

Hence if $\tilde{L}$ is a polynomial of degree $P$ the constants in Assumption 1 will take the following values $B_{\tilde{L}} = d^{P/2} \left(\sum_{\alpha,|\alpha| \le P} |A_\alpha|^2\right)^{1/2}$, and $r = P$.

Finally, since we are using an approximation of the function $L$ we will incur an error at each step of the iteration. The following Lemma shows that the error between the iterates $u_t$ and the approximate iterates $\tilde{u}_t$ increases with $t$. The error is calculated by recursively tracking the error between $u_t$ and $\tilde{u}_t$ for each $t$ in terms of the error at $t-1$. Note that this error can be controlled by using smaller values of $\eta$.

**Lemma 8.** *Let $\tilde{L} : \mathbb{R}^d \to \mathbb{R}$ be the function satisfying the properties in Assumption 1 such that $\sup_{x \in \mathbb{R}^d} \|\nabla L(x) - \nabla \tilde{L}(x)\|_2 \le \epsilon_L \|x\|_2$ and we have $\mathcal{E}(u) = \int_\Omega L(\nabla u)dx$ and $\tilde{\mathcal{E}}(u) = \int_\Omega \tilde{L}(\nabla u)dx$. For $\eta \in (0, \frac{\lambda^3}{(C_p+1)\Lambda^3}]$ consider the sequences,*

$$u_{t+1} = u_t - \eta(I - \Delta)^{-1}D\mathcal{E}(u_t), \text{ and } \tilde{u}_{t+1} = \tilde{u}_t - \eta(I - \Delta)^{-1}D\tilde{\mathcal{E}}(u_t)$$

*then for all $t \in \mathbb{N}$ and $R \le \|u^\star\|_{H_0^1(\Omega)} + \frac{1}{\lambda}\mathcal{E}(u_0)$ we have,*

$$\|u_t - \tilde{u}_t\|_{H_0^1(\Omega)} \le \frac{\epsilon_L \eta R}{\Lambda + \epsilon_L} \left((1 + \eta(\Lambda + \epsilon_L))^t - 1\right)$$

## 7 CONCLUSION AND FUTURE WORK

In this work, we take a representational complexity perspective on neural networks, as they are used to approximate solutions of nonlinear variational elliptic PDEs of the form $-\text{div}(\nabla L(\nabla u)) = 0$. We prove that if $L$ is such that composing $L$ with function of bounded Barron norm increases the Barron norm in a bounded fashion, then we can bound the Barron norm of the solution $u^\star$ to the PDE—potentially evading the curse of dimensionality depending on the rate of this increase. Our results subsume and vastly generalize prior work on the linear case (Marwah et al., 2021; Chen et al., 2021). Our proof consists of neurally simulating preconditioned gradient descent on the energy function defining the PDE, which we prove is strongly convex in an appropriate sense.

There are many potential avenues for future work. Our techniques (and prior techniques) strongly rely on the existence of a variational principle characterizing the solution of the PDE. In classical PDE literature, these classes of PDEs are also considered better behaved: e.g., proving regularity bounds is much easier for such PDEs (Fernández-Real & Ros-Oton, 2020). There are many nonlinear PDEs that come without a variational formulation—e.g. the Monge-Ampere equation—for which regularity estimates are derived using non-constructive methods like comparison principles. It is a wide-open question to construct representational bounds for any interesting family of PDEs of this kind. It is also a very interesting question to explore other notions of complexity—e.g. number of parameters in a (potentially deep) network like in Marwah et al. (2021), Rademacher complexity, among others.

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

# A   APPENDIX

## A.1   PROOF FOR LEMMA 1

The proof follows form  Fernández-Real & Ros-Oton (2020) Chapter 3. We are provided it here for completeness.

*Proof of Lemma 1.* If the function $u^\star$ minimizes the energy functional in Definition 1 then we have for all $\epsilon \in \mathbb{R}$

$$\mathcal{E}(u) \leq \mathcal{E}(u + \epsilon\varphi)$$

where $\varphi \in C_c^\infty(\Omega)$. That is, we have a minima at $\epsilon = 0$ and taking a derivative w.r.t $\epsilon$ we get,

$$
\begin{aligned}
d\mathcal{E}[u](\varphi) &= \lim_{\epsilon \to 0} \frac{\mathcal{E}(u + \epsilon\varphi) - \mathcal{E}(u)}{\epsilon} = 0 \\
&= \lim_{\epsilon \to 0} \frac{\int_\Omega L\left(\nabla u + \epsilon\nabla\varphi\right)dx - \int_\Omega L(\nabla u)dx}{\epsilon} = 0 \\
&= \lim_{\epsilon \to 0} \frac{\int_\Omega \epsilon\nabla L(\nabla u)\nabla\varphi + \frac{\epsilon^2}{2}r(x)}{\epsilon} = 0
\end{aligned}
$$

where for all $x \in \Omega$ we have $|r(x)| \leq |\nabla\varphi|^2 \sup_{p \in \mathbb{R}^d} D^2 L(p)$.

Since $\epsilon \to 0$ the final derivative is of the form,

$$d\mathcal{E}[u](\varphi) = \int_\Omega \nabla L(\nabla u)\nabla\varphi dx = 0 \tag{10}$$

For functions $r, s \in H_0^1(\Omega)$ note the following Green's formula,

$$
\begin{aligned}
\int_\Omega (\nabla r)s dx &= -\int_\Omega \nabla r \cdot \nabla s dx + \int_{\partial\Omega} \frac{\partial u}{\partial n}v d\Gamma \\
&\implies \int_\Omega (\nabla r)s dx = -\int_\Omega \nabla r \cdot \nabla s dx
\end{aligned}
\tag{11}
$$

Using the identity in Equation 11 in Equation 10 we get,

$$d\mathcal{E}[u](\varphi) = \int_\Omega -\text{div}\left(\nabla L(\nabla u)\right)\varphi dx = 0$$

That is the minima for the energy functional is reached at a $u$ which solves the following PDE,

$$d\mathcal{E}(u) = -\text{div}\left(\nabla L(\nabla u)\right) = 0.$$

$\square$

# B   PROOFS FROM SECTION 6.1

## B.1   PROOF FOR LEMMA 2

*Proof.* In order to prove part 1, we will use the following integration by parts identity, for functions $r : \Omega \to \mathbb{R}$ such that and $s : \Omega \to \mathbb{R}$, and $r, s \in H_0^1(\Omega)$,

$$\int_\Omega \frac{\partial r}{\partial x_i}s dx = -\int_\Omega r\frac{\partial s}{\partial x_i}dx + \int_{\partial\Omega} rsn d\Gamma \tag{12}$$

where $n_i$ is a normal at the boundary and $d\Gamma$ is an infinitesimal element of the boundary $\partial\Omega$.

Using the formula in Equation 12 for functions $u, v \in H_0^1(\Omega)$, we have

$$
\begin{aligned}
\langle D\mathcal{E}(u), v\rangle_{L^2(\Omega)} &= \langle -\nabla \cdot \nabla L(\nabla u), v\rangle_{L^2(\Omega)} \\
&= -\int_\Omega \nabla \cdot \nabla L(\nabla u) v dx \\
&= -\int_\Omega \sum_{i=1}^d \frac{\partial (\nabla L(\nabla u))_i}{\partial x_i} v dx \\
&= \int_\Omega \sum_{i=1}^d (\nabla L(\nabla u))_i \frac{\partial v}{\partial x_i} dx + \int_\Omega \sum_{i=1}^d (\nabla L(\nabla u))_i v n_i dx \\
&= \int \nabla L(\nabla u) \cdot \nabla v dx
\end{aligned}
$$

where in the last equality we use the fact that the function $v \in H_0^1(\Omega)$, thus $v(x) = 0, \forall x \in \partial\Omega$.

To show the second part, first note since $L : \mathbb{R}^d \to \mathbb{R}$ is strongly convex and smooth, we have

$$
\forall x, y \in \mathbb{R}^d, \lambda\|x - y\|_2 \le \|\nabla L(x) - \nabla L(y)\|_2 \le \Lambda\|x - y\|_2. \tag{13}
$$

This implies

$$
\forall x \in \Omega, \|\nabla L(\nabla u(x)) - \nabla L(\nabla v(x))\|_2 \le \Lambda\|\nabla u(x) - \nabla v(x)\|_2 \tag{14}
$$

Taking square on each side and itegrating over $\Omega$ we get

$$
\begin{aligned}
\int_\Omega \|\nabla L(\nabla u(x)) - \nabla L(\nabla v(x))\|_2^2 dx &\le \Lambda^2 \int_\Omega \|\nabla u(x) - \nabla v(x)\|_2^2 dx \\
\implies \|\nabla L(\nabla u) - \nabla L(\nabla v)\|_{L^2(\Omega)} &\le \Lambda\|\nabla u - \nabla v\|_{L^2(\Omega)}
\end{aligned}
$$

On the other hand, from part 1, we have

$$
\langle D\mathcal{E}(u) - D\mathcal{E}(v), u - v\rangle_{L^2(\Omega)} = \langle \nabla L(\nabla u) - \nabla L(\nabla v), \nabla u - \nabla v\rangle_{L^2(\Omega)}
$$

Hence, by Cauchy-Schwartz, we get

$$
\begin{aligned}
\langle D\mathcal{E}(u) - D\mathcal{E}(v), u - v\rangle_{L^2(\Omega)} &= \langle \nabla L(\nabla u) - \nabla L(\nabla v), \nabla u - \nabla v\rangle_{L^2(\Omega)} \\
&\le \|\nabla L(\nabla u) - \nabla L(\nabla v)\|_{L^2(\Omega)} \|\nabla u - \nabla v\|_{L^2(\Omega)} \\
&\le \Lambda\|\nabla u - \nabla v\|_{L^2(\Omega)}^2
\end{aligned}
$$

which proves the right hand side of the inequality in part 2.

For the left size of the inequality, by convexity of $L$ we have $\forall x, y \in \mathbb{R}^d, (\nabla L(x) - \nabla L(y))^T(x - y) \ge \lambda\|x - y\|_2^2$ is convex,

$$
(\nabla L(\nabla u(x)) - \nabla L(\nabla v(x)))^T (\nabla u(x) - \nabla v(x)) \ge \lambda\|\nabla u(x) - \nabla v(x)\|_2^2
$$

Integrating over $\Omega$ we get,

$$
\begin{aligned}
\int_\Omega (\nabla L(\nabla u(x)) - \nabla L(\nabla v(x)))^T (\nabla u(x) - \nabla v(x)) dx &\ge \lambda \int_\Omega \|\nabla u(x) - \nabla v(x)\|_2^2 dx \\
\implies \langle \nabla L(\nabla u) - \nabla L(\nabla v), \nabla u - \nabla v\rangle_{L^2(\Omega)} &\ge \lambda\|\nabla u - \nabla v\|_{L^2(\Omega)}^2
\end{aligned} \tag{15}
$$

Using part 1 again, this implies Equation 15

$$
\begin{aligned}
\langle D\mathcal{E}(u) - D\mathcal{E}(v), u - v\rangle_{L^2(\Omega)} &= \langle \nabla L(\nabla u) - \nabla L(\nabla v), \nabla u - \nabla v\rangle_{L^2(\Omega)} \\
&\ge \lambda\|\nabla v - \nabla v\|_{L^2(\Omega)}^2.
\end{aligned}
$$

as we wanted.

To show part 3, we first Taylor expand $L$ to rewrite the energy function as:

$$\mathcal{E}(u + v) = \int_\Omega L(\nabla u(x) + \nabla v(x))dx$$
$$= \int_\Omega \left( L(\nabla u(x)) + \nabla L(\nabla u(x))\nabla v(x) + \frac{1}{2}D^2 L(\tilde{x})\|\nabla v(x)\|_2^2 \right) dx \tag{16}$$

where $\tilde{x} \in \mathbb{R}^d$ (and is potentially different for every $x \in \Omega$). Since the function $L$ is strongly convex we have

$$\lambda I_d \leq D^2 L(\tilde{x}) \leq \Lambda I_d.$$

Plugging in these bounds in Equation 16, we have:

$$\mathcal{E}(u + v) \leq \int_\Omega \left( L(\nabla u(x)) + \nabla L(\nabla u(x))\nabla v(x) + \frac{\Lambda}{2}\|\nabla v(x)\|^2 \right) dx$$
$$\implies \mathcal{E}(u + v) \leq \mathcal{E}(u) + \langle D\mathcal{E}(u), v \rangle_{L^2(\Omega)} + \frac{\Lambda}{2}\langle \nabla v, \nabla v \rangle_{L^2(\Omega)}. \tag{17}$$

as well as

$$\mathcal{E}(u + v) \geq \int_\Omega \left( L(\nabla u(x)) + \nabla L(\nabla u(x))\nabla v(x) + \frac{\Lambda}{2}\|\nabla v(x)\|^2 \right) dx$$
$$\implies \mathcal{E}(u + v) \geq \mathcal{E}(u) + \langle D\mathcal{E}(u), v \rangle_{L^2(\Omega)} + \frac{\lambda}{2}\langle \nabla v, \nabla v \rangle_{L^2(\Omega)}. \tag{18}$$

Combining Equation 17 and Equation 18 we get,

$$\frac{\lambda}{2}\|\nabla v\|_{L^2(\Omega)}^2 + \langle D\mathcal{E}(u), v \rangle_{L^2(\Omega)} \leq \mathcal{E}(u + v) - \mathcal{E}(u) \leq \langle D\mathcal{E}(u), v \rangle_{L^2(\Omega)} + \frac{\Lambda}{2}\|\nabla v\|_{L^2(\Omega)}^2$$

Finally, part 4 follows by plugging in $u = u^\star$ and $v = u - u^\star$ in part 3 and using the fact that $D\mathcal{E}(u^\star) = 0$. $\qquad\square$

### B.2  PROOF FOR LEMMA 3

*Proof.* Let $\{\lambda_i, \phi_i\}_{i=1}^\infty$ denote the (eigenvalue, eigenfunction) pairs of the operator $-\Delta$ where $0 < \lambda_1 \leq \lambda_2 \leq \cdots$, which are real and countable. ( Evans (2010), Theorem 1, Section 6.5)

Using the definition of eigenvalues and eigenfunctions, we have

$$\lambda_1 = \inf_{v \in H_0^1(\Omega)} \frac{\langle -\Delta v, v \rangle_{L^2(\Omega)}}{\|v\|_{L^2(\Omega)}^2}$$
$$= \inf_{v \in H_0^1(\Omega)} \frac{\langle \nabla v, \nabla v \rangle_{L^2(\Omega)}}{\|v\|_{L^2(\Omega)}^2}$$
$$= \frac{1}{C_p}.$$

where in the last equality we use Theorem 2.

Let us write the functions $v, w$ in the eigenbasis as $v = \sum_i \mu_i \phi_i$. Notice that an eigenfunction of $-\Delta$ is also an eigenfunction for $(I - \Delta)^{-1}$, with correspondinding eigenvalue $\frac{1}{1+\lambda_i}$.

Thus, to show part 1, we have,

$$\left\|(I-\Delta)^{-1}\nabla\cdot\nabla v\right\|^2_{L^2(\Omega)} = \left\|(I-\Delta)^{-1}\Delta v\right\|^2_{L^2(\Omega)}$$

$$= \left\|\sum_{i=1}^\infty \frac{\lambda_i}{1+\lambda_i}\mu_i\phi_i\right\|^2_{L^2(\Omega)}$$

$$\leq \left\|\sum_{i=1}^\infty \mu_i\phi_i\right\|^2_{L^2(\Omega)}$$

$$= \sum_{i=1}^\infty \mu_i^2 = \|u\|^2_{L^2(\Omega)}$$

where in the last equality we use the fact that $\phi_i$ are orthogonal.

Now, note that $(I-\Delta)^{-1}v = \sum_{i=1}^\infty \frac{1}{(1+\lambda_i)}\mu_i\phi_i$ now, note that since $\lambda_1 \leq \lambda_2 \leq \cdots$ and $\frac{x}{1+x}$ is monotonically increasing, we have for all $i \in \mathbb{N}$

$$\frac{1}{1+\lambda_i} \geq \frac{1}{(1+C_p)\lambda_i} \tag{19}$$

and note that $\frac{1}{\lambda_i}$ are the eigenvalues for $(-\Delta)^{-1}$ for all $i \in \mathbb{N}$.

Now, bounding $\langle(I-\Delta)^{-1}v, v\rangle_{L^2(\Omega)}$

$$\langle(I-\Delta)^{-1}v, v\rangle_{L^2(\Omega)} = \left\langle \sum_{i=1}^\infty \frac{\mu_i}{1+\lambda_i}\phi_i, \sum_{i=1}^\infty \mu_i\phi_i \right\rangle_{L^2(\Omega)}$$

$$= \sum_{i=1}^\infty \frac{\mu_i^2}{1+\lambda_i}\|\phi_i\|^2_{L^2(\Omega)} \tag{20}$$

where we the orthogonality of $\phi_i's$ to get Equation 20. Using the inequality in Equation 19 we can further lower bound $\langle(I-\Delta)^{-1}v, v\rangle_{L^2(\Omega)}$ as follows,

$$\langle(I-\Delta)^{-1}v, v\rangle_{L^2(\Omega)} \geq \sum_{i=1}^\infty \frac{\mu_i^2}{(1+C_p)\lambda_i}\|\phi_i\|^2_{L^2(\Omega)} := \frac{1}{1+C_p}\langle(-\Delta)^{-1}v, v\rangle_{L^2(\Omega)},$$

where we use the following set of equalities in the last step,

$$\langle(-\Delta)^{-1}v, v\rangle_{L^2(\Omega)} = \left\langle \sum_{i=1}^\infty \frac{\mu_i}{\lambda_i}\phi_i, \sum_{i=1}^\infty \mu_i\phi_i \right\rangle_{L^2(\Omega)} = \sum_{i=1}^\infty \frac{\mu_i^2}{\lambda_i}\|\phi_i\|^2_{L^2(\Omega)}. \qquad \square$$

### B.3 Proof for Convergence: Proof for Lemma 4

*Proof.* For the analysis we consider $\eta = \frac{\lambda^3}{(1+C_p)\Lambda^3}$

Taylor expanding as in Equation 17, we have

$$\mathcal{E}(u_{t+1}) \leq \mathcal{E}(u_t) - \eta \underbrace{\left\langle \nabla L(\nabla u_t), \nabla(I-\Delta)^{-1}D\mathcal{E}(u_t) \right\rangle_{L^2(\Omega)}}_{\text{Term 1}}$$

$$+ \underbrace{\frac{\eta^2\Lambda^2}{2}\left\|\nabla(I-\Delta)^{-1}D\mathcal{E}(u_t)\right\|^2_{L^2(\Omega)}}_{\text{Term 2}}. \tag{21}$$

where we have in Equation 17 plugged in $u_{t+1} - u_t = -\eta(I-\Delta)^{-1}D\mathcal{E}(u_t)$.

First we lower bound *Term 1*. Since $u^\star$ is the solution to the PDE in Equation 3, we have $D\mathcal{E}(u^\star) = 0$. Therefore we have

$$\left\langle \nabla L(\nabla u_t), \nabla(I-\Delta)^{-1}D\mathcal{E}(u_t) \right\rangle_{L^2(\Omega)} = \left\langle \nabla L(\nabla u_t), \nabla(I-\Delta)^{-1}(D\mathcal{E}(u_t) - D\mathcal{E}(u^\star)) \right\rangle_{L^2(\Omega)} \tag{22}$$

Similarly, since $u^\star$ is the solution to the PDE in Equation 3 Equation 2 we have for all $\varphi \in H_0^1(\Omega)$ $\langle \nabla L(\nabla u^\star), \nabla \varphi \rangle_{L^2(\Omega)} = 0$. Using this Equation 22 we get,

$$\left\langle \nabla L(\nabla u_t), \nabla(I - \Delta)^{-1} D\mathcal{E}(u_t) \right\rangle_{L^2(\Omega)}$$

$$= \left\langle \nabla L(\nabla u_t), \nabla(I - \Delta)^{-1} \left( D\mathcal{E}(u_t) - D\mathcal{E}(u^\star) \right) \right\rangle_{L^2(\Omega)}$$

$$= \left\langle \nabla L(\nabla u_t), \nabla(I - \Delta)^{-1} \left( D\mathcal{E}(u_t) - D\mathcal{E}(u^\star) \right) \right\rangle_{L^2(\Omega)} + \left\langle \nabla L(\nabla u^\star), \nabla(I - \Delta)^{-1} \left( D\mathcal{E}(u_t) - D\mathcal{E}(u^\star) \right) \right\rangle_{L^2(\Omega)}$$

$$= \left\langle \nabla L(\nabla u_t) - \nabla L(\nabla u^\star), \nabla(I - \Delta)^{-1} \left( D\mathcal{E}(u_t) - D\mathcal{E}(u^\star) \right) \right\rangle_{L^2(\Omega)} \tag{23}$$

Using Equation 23, we can rewrite Term 1 as

$$\left\langle \nabla L(\nabla u_t), \nabla(I - \Delta)^{-1} D\mathcal{E}(u_t) \right\rangle_{L^2(\Omega)}$$

$$= \left\langle \nabla L(\nabla u_t) - \nabla L(\nabla u^\star), \nabla(I - \Delta)^{-1} \left( D\mathcal{E}(u_t) - D\mathcal{E}(u^\star) \right) \right\rangle_{L^2(\Omega)}$$

$$= \int_\Omega \left( \nabla L(\nabla u_t) - \nabla L(\nabla u^\star) \right) \cdot \nabla(I - \Delta)^{-1} \left( -\nabla \cdot (\nabla L(\nabla u_t)) - \nabla L(u^\star) \right) dx$$

$$\overset{(i)}{=} \int_\Omega \left( \nabla L(\nabla u_t) - \nabla L(\nabla u^\star) \right) \cdot \nabla(I - \Delta)^{-1} \left( -\Delta \left( L(\nabla u_t) - L(u^\star) \right) \right) dx$$

$$= \int_\Omega \left( \nabla L(\nabla u_t) - \nabla L(\nabla u^\star) \right) \cdot (I - \Delta)^{-1}(-\Delta) \left( \nabla L(\nabla u_t) - \nabla L(u^\star) \right) dx \tag{24}$$

where in step $(i)$ we use $-\nabla \cdot \nabla v = -\Delta v$ and the fact that $\nabla$ commutes with $(I - \Delta)^{-1}(-\Delta)$.

Plugging *part 2* of Lemma 3 in Equation 24, we get

$$\left\langle \nabla L(\nabla u_t), \nabla(I - \Delta)^{-1} D\mathcal{E}(u_t) \right\rangle_{L^2(\Omega)}$$

$$\geq \frac{1}{1 + C_p} \int_\Omega \left( \nabla L(\nabla u_t) - \nabla L(\nabla u^\star) \right) \cdot (-\Delta)^{-1}(-\Delta) \left( \nabla L(\nabla u_t) - \nabla L(\nabla u^\star) \right) dx$$

$$\geq \frac{1}{1 + C_p} \left\langle \nabla L(\nabla u_t) - \nabla L(\nabla u^\star), \nabla L(\nabla u_t) - \nabla L(\nabla u^\star) \right\rangle_{L^2(\Omega)}$$

$$\overset{(i)}{\geq} \frac{\lambda^2}{1 + C_p} \|\nabla u_t - \nabla u^\star\|_{L^2(\Omega)}^2$$

$$\geq \frac{2\lambda^2}{(1 + C_p)\Lambda} \left( \mathcal{E}(u_t) - \mathcal{E}(u^\star) \right) \tag{25}$$

where $(i)$ follows by part 2 of Lemma 2 and we use part 4 of Lemma 2 for the last inequality.

We will proceed to upper bounding *Term 2*. Using part 1 of Lemma 3 we have

$$\left\| \nabla(I - \Delta)^{-1} D\mathcal{E}(u_t) \right\|_{L^2(\Omega)}^2 = \left\| \nabla(I - \Delta)^{-1} \left( D\mathcal{E}(u_t) - D\mathcal{E}(u^\star) \right) \right\|_{L^2(\Omega)}^2$$

$$= \left\| \nabla(I - \Delta)^{-1} \left( -\nabla \cdot (\nabla L(\nabla u_t) - \nabla L(\nabla u^\star)) \right) \right\|_{L^2(\Omega)}^2$$

$$\overset{(i)}{=} \left\| \nabla(I - \Delta)^{-1}(-\Delta) \left( L(\nabla u_t) - L(\nabla u^\star) \right) \right\|_{L^2(\Omega)}^2$$

$$\overset{(ii)}{=} \left\| (I - \Delta)^{-1}(-\Delta) \left( \nabla L(\nabla u_t) - \nabla L(\nabla u^\star) \right) \right\|_{L^2(\Omega)}^2$$

$$\overset{(iii)}{\leq} \left\| \nabla L(\nabla u_t) - \nabla L(\nabla u^\star) \right\|_{L^2(\Omega)}^2$$

$$\leq \Lambda^2 \|\nabla u_t - \nabla u^\star\|_{L^2(\Omega)}^2$$

$$\leq \frac{2\Lambda^2}{\lambda} \left( \mathcal{E}(u_t) - \mathcal{E}(u^\star) \right). \tag{26}$$

Here, we use the fact that $-\nabla \cdot \nabla = -\Delta$ in step $(i)$ and the fact that $\nabla$ commutes with $(I - \Delta)^{-1}(-\Delta)$ in step $(ii)$, and finally we use the result from part 2 of Lemma 3 to get the inequality in $(iii)$.

Combining Equation 25 and Equation 26 in Equation 21 we get

$$\implies \mathcal{E}(u_{t+1}) - \mathcal{E}(u^\star) \le \mathcal{E}(u_t) - \mathcal{E}(u^\star) - \left( \frac{2\lambda^2}{(1+C_p)\Lambda} - \eta\frac{\Lambda^2}{\lambda} \right) \eta \left( \mathcal{E}(u_t) - \mathcal{E}(u^\star) \right)$$

Since $\eta = \lambda^3/((1+C_p)\Lambda^3)$ we have

$$\mathcal{E}(u_{t+1}) - \mathcal{E}(u^\star) \le \mathcal{E}(u_t) - \mathcal{E}(u^\star) - \frac{\lambda^5}{(1+C_p)\Lambda^4}\eta\left(\mathcal{E}(u_t) - \mathcal{E}(u^\star)\right)$$

$$\implies \mathcal{E}(u_{t+1}) - \mathcal{E}(u^\star) \le \left( 1 - \frac{\lambda^5}{(1+C_p)\Lambda^4} \right)^t \left( \mathcal{E}(u_0) - \mathcal{E}(u^\star) \right). \qquad \square$$

## C    ERROR ANALYSIS

First, we will need the following simple technical lemma showing that the $H_0^1(\Omega)$ norm is self-dual:

**Lemma 9.** *The dual norm of $\| \cdot \|_{H_0^1(\Omega)}$ is $\| \cdot \|_{H_0^1(\Omega)}$.*

*Proof.* If $\|u\|_*$ denotes the dual norm of $\|u\|_{H_0^1(\Omega)}$, by definition we have,

$$
\begin{aligned}
\|u\|_* &= \sup_{\substack{v \in H_0^1(\Omega) \\ \|v\|_{H_0^1(\Omega)}=1}} \langle u, v \rangle_{H_0^1(\Omega)} \\
&= \sup_{\substack{v \in H_0^1(\Omega) \\ \|v\|_{H_0^1(\Omega)}=1}} \langle \nabla u, \nabla v \rangle_{L^2(\Omega)} \\
&\le \sup_{\substack{v \in H_0^1(\Omega) \\ \|v\|_{H_0^1(\Omega)}=1}} \|\nabla u\|_{L^2(\Omega)} \|\nabla v\|_{L^2(\Omega)} \\
&= \|\nabla u\|_{L^2(\Omega)}
\end{aligned}
$$

where the inequality follows by Cauchy- Schwarz. On the other hand, equality can be achieved by taking $v = \frac{u}{\|\nabla u\|_2}$. Thus, $\|u\|_* = \|\nabla u\|_{L^2(\Omega)} = \|u\|_{H_0^1(\Omega)}$ as we wanted.    $\square$

With this we can prooceed to the proof of Lemma 8.

### C.1    PROOF FOR LEMMA 8

*Proof.* We define for all $t$ $r_t = \tilde{u}_t - u_t$, and will iteratively bound $\|r_t\|_{L^2(\Omega)}$.

Starting with $u_0 = 0$ and $\tilde{u}_t = 0$, we define the iterative sequences as,

$$\begin{cases} u_0 = u_0 \\ u_{t+1} = u_t - \eta(I - \Delta)^{-1}D\mathcal{E}(u_t) \end{cases}$$

$$\begin{cases} \tilde{u}_t = u_0 \\ \tilde{u}_{t+1} = \tilde{u}_t - \eta(I - \Delta)^{-1}D\tilde{\mathcal{E}}(\tilde{u}_t) \end{cases}$$

where $\eta \in (0, \frac{\lambda^3}{(1+C_p)\Lambda^3}]$. Subtracting the two we get,

$$\tilde{u}_{t+1} - u_{t+1} = \tilde{u}_t - u_t - \eta(I - \Delta)^{-1}\left( D\tilde{\mathcal{E}}(\tilde{u}_t) - D\mathcal{E}(u_t) \right)$$

$$\implies r_{t+1} = r_t - \eta(I - \Delta)^{-1}\left( D\tilde{\mathcal{E}}(u_t + r_t) - D\mathcal{E}(u_t) \right) \qquad (27)$$

Taking $H_0^1(\Omega)$ norm on both sides we get,

$$\|r_{t+1}\|_{H_0^1(\Omega)} \le \|r_t\|_{H_0^1(\Omega)} + \eta \left\| (I - \Delta)^{-1}\left( D\tilde{\mathcal{E}}(u_t + r_t) - D\mathcal{E}(u_t) \right) \right\|_{H_0^1(\Omega)} \qquad (28)$$

Towards bounding $\left\|(I-\Delta)^{-1}D\tilde{\mathcal{E}}(u_t + r_t) - D\mathcal{E}(u_t)\right\|_{H_0^1(\Omega)}$, from Lemma 9 we know that the dual norm of $\|w\|_{H_0^1(\Omega)}$ is $\|w\|_{H_0^1(\Omega)}$, thus,

$$\left\|(I-\Delta)^{-1}D\tilde{\mathcal{E}}(u_t + r_t) - D\mathcal{E}(u_t)\right\|_{H_0^1(\Omega)}$$

$$= \sup_{\substack{\varphi \in H_0^1(\Omega) \\ \|\varphi\|_{H_0^1(\Omega)}=1}} \left\langle \nabla(I-\Delta)^{-1}\left(D\tilde{\mathcal{E}}(u_t + r_t) - D\mathcal{E}(u_t)\right), \nabla\varphi \right\rangle_{L^2(\Omega)}$$

$$= \sup_{\substack{\varphi \in H_0^1(\Omega) \\ \|\varphi\|_{H_0^1(\Omega)}=1}} \left\langle \nabla(I-\Delta)^{-1}\left(D\tilde{\mathcal{E}}(u_t + r_t) - D\mathcal{E}(u_t + r_t)\right), \nabla\varphi \right\rangle_{L^2(\Omega)}$$

$$+ \sup_{\substack{\varphi \in H_0^1(\Omega) \\ \|\varphi\|_{H_0^1(\Omega)}=1}} \left\langle \nabla(I-\Delta)^{-1}\left(D\mathcal{E}(u_t + r_t) - D\mathcal{E}(u_t)\right), \nabla\varphi \right\rangle_{L^2(\Omega)}$$

$$= \sup_{\substack{\varphi \in H_0^1(\Omega) \\ \|\varphi\|_{H_0^1(\Omega)}=1}} \left\langle \nabla(I-\Delta)^{-1}\nabla \cdot \left(\nabla\tilde{L}(\nabla u_t + \nabla r_t) - \nabla L(\nabla u_t + \nabla r_t)\right), \nabla\varphi \right\rangle_{L^2(\Omega)}$$

$$+ \sup_{\substack{\varphi \in H_0^1(\Omega) \\ \|\varphi\|_{H_0^1(\Omega)}=1}} \left\langle \nabla(I-\Delta)^{-1}\nabla \cdot (\nabla L(\nabla u_t + \nabla r_t)) - \nabla L(\nabla u_t), \nabla\varphi \right\rangle_{L^2(\Omega)}$$

$$\leq \sup_{\substack{\varphi \in H_0^1(\Omega) \\ \|\varphi\|_{H_0^1(\Omega)}=1}} \left\langle \nabla\tilde{L}(\nabla u_t + \nabla r_t) - \nabla L(\nabla u_t + \nabla r_t), \nabla\varphi \right\rangle_{L^2(\Omega)}$$

$$+ \sup_{\substack{\varphi \in H_0^1(\Omega) \\ \|\varphi\|_{H_0^1(\Omega)}=1}} \left\langle \nabla L(\nabla u_t + \nabla r_t) - \nabla L(\nabla u_t), \nabla\varphi \right\rangle_{L^2(\Omega)}$$

$$\leq \epsilon_L \|\nabla u_t + \nabla r_t\|_{L^2(\Omega)} + \|\nabla L(\nabla u_t + \nabla r_t) - \nabla L(\nabla u_t)\|_{L^2(\Omega)} \tag{29}$$

Using the Lipschitzness of $\nabla L$, we have

$$\|\nabla L(\nabla u_t(x) + \nabla r_t(x)) - \nabla L(\nabla u_t)\|_2 \leq \|\nabla L(\nabla u_t(x)) - \nabla L(\nabla u_t(x))\|_2 + \sup_{p \in \mathbb{R}^d} D^2 F(p)\|\nabla r_t(x)\|_2$$

$$\leq \Lambda\|\nabla r_t(x)\|_2 \tag{30}$$

Squaring and integrating over $\Omega$ on both sides we get

$$\|\nabla L(\nabla u_t(x) + \nabla r_t(x)) - \nabla L(\nabla u_t)\|_{L^2(\Omega)} \leq \Lambda\|\nabla r_t\|_{L^2(\Omega)} \tag{31}$$

Pluggin in Equation 31 in Equation 29 we get,

$$\left\|(I-\Delta)^{-1}D\tilde{\mathcal{E}}(u_t + r_t) - D\mathcal{E}(u_t)\right\|_{H_0^1(\Omega)} \leq (\Lambda + \epsilon_L)\|\nabla r_t\|_{L^2(\Omega)} + \epsilon_L\|\nabla u_t\|_{L^2(\Omega)} \tag{32}$$

Furthermore, from Lemma 4 we have for all $t \in \mathbb{N}$,

$$\mathcal{E}(u_t) - \mathcal{E}(u^\star) \leq \left(1 - \frac{\lambda^5}{C_p\Lambda^4}\right)^t \mathcal{E}(u_0)$$

$$\leq \mathcal{E}(u_0)$$

and

$$\|u_t - u^\star\|_{H_0^1(\Omega)} \leq \frac{2}{\lambda}\left(\mathcal{E}(u_t) - \mathcal{E}(u_0)\right)$$

$$\leq \frac{2}{\lambda}\mathcal{E}(u_0)$$

Hence we have that for all $t \in \mathbb{N}$,

$$\|u_t\|_{H_0^1(\Omega)} \leq \|u^\star\|_{H_0^1(\Omega)} + \frac{2}{\lambda}\mathcal{E}(u_0) =: R.$$

Putting this all together, we have

$$\left\| (I - \Delta)^{-1} D\tilde{\mathcal{E}}(u_t + r_t) - D\mathcal{E}(u_t) \right\|_{H_0^1(\Omega)} \leq (\Lambda + \epsilon_L) \|\nabla r_t\|_{L^2(\Omega)} + \epsilon_L R \tag{33}$$

Hence using the result from Equation 33 in Equation 28 to get,

$$\|r_{t+1}\|_{H_0^1(\Omega)} \leq (1 + \eta(\Lambda + \epsilon_L)) \|r_t\|_{H_0^1(\Omega)} + \epsilon_L \eta R$$

$$\implies \|r_{t+1}\|_{H_0^1(\Omega)} \leq \frac{\epsilon_L \eta R}{\Lambda + \epsilon_L} \left( (1 + \eta(\Lambda + \epsilon_L))^t - 1 \right) \tag{34}$$

where we use the fact that $\|r_t\|_{H_0^1(\Omega)} = 0$.

Notice that $\epsilon_L << \Lambda$. Further, we have $\eta \in (0, \frac{\lambda^3}{(C_p+1)\Lambda^3}]$ that is $\eta \leq 1$, it implies that $\eta(\Lambda + \epsilon_L) < 1$.

Hence we can further bound Equation 34 as follows,

$$\|r_{t+1}\|_{H_0^1(\Omega)} \leq \frac{\epsilon_L \eta R}{\Lambda + \epsilon_L} \left( (1 + \eta(\Lambda + \epsilon_L))^t - 1 \right)$$

$$\square$$

## D  PROOFS FOR BARRON NORM APPROXIMATION: SECTION 6.2

### D.1  PROOF FOR LEMMA 5: BARRON NORM RECURSION

*Proof.* Note that the update equation looks like,

$$\begin{aligned} u_{t+1} &= u_t - \eta(I - \Delta)^{-1} D\mathcal{E}(u_t) \\ &= u_t - \eta(I - \Delta)^{-1} \left( -\nabla \cdot \nabla L(\nabla u_t) \right) \\ &= u_t - \eta(I - \Delta)^{-1} \left( -\sum_{i=1}^d \partial_i^2 L(\nabla u_t) \right) \end{aligned} \tag{35}$$

From Lemma 6 we have

$$\|\nabla u_t\|_{\mathcal{B}(\Omega)} = \max_{i \in [d]} \|\partial_i u_t\|_{\mathcal{B}(\Omega)} \leq W_t \|u_t\|_{\mathcal{B}(\Omega)} \tag{36}$$

Note that since $u_t \in \Gamma_{W_t}$ we have $\nabla u_t \in \Gamma_{W_t}$ and $L(\nabla u_t) \in \Gamma_{kW_t}$ (from Assumption 1).

Therefore, we can bound the Barron norm as,

$$\begin{aligned} \left\| (I - \Delta)^{-1} \left( -\sum_{i=1}^d \partial_i^2 L(\nabla u_t) \right) \right\|_{\mathcal{B}(\Omega)} &\overset{(i)}{\leq} \left\| -\sum_{i=1}^d \partial_i^2 L(\nabla u_t) \right\|_{\mathcal{B}(\Omega)} \\ &\overset{(ii)}{\leq} d \left\| \partial_i^2 L(\nabla u_t) \right\|_{\mathcal{B}(\Omega)} \\ &\leq d(kW_t)^2 \|L(\nabla u_t)\|_{\mathcal{B}(\Omega)} \\ &\leq d(kW_t)^2 B_{\tilde{L}} \|u_t\|_{\mathcal{B}(\Omega)}^r \end{aligned}$$

where we use the fact that for a function $h$, we have $\|(I - \Delta)^{-1} h\|_{\mathcal{B}(\Omega)} \leq \|h\|_{\mathcal{B}(\Omega)}$ from Lemma 6 in $(i)$ and the fact that $L(\nabla u_t) \in \Gamma_{kW_t}$ in $(ii)$. Using the result of *Addition* from Lemma 6 we have

$$\begin{aligned} \|u_t\|_{\mathcal{B}(\Omega)} &\leq \|u_t\|_{\mathcal{B}(\Omega)} + \left( \eta d(kW_t)^2 B_{\tilde{L}} \right) \|u_t\|_{\mathcal{B}(\Omega)}^r \\ &\leq \left( 1 + \eta d(kW_t)^2 B_{\tilde{L}} \right) \|u_t\|_{\mathcal{B}(\Omega)}^r. \end{aligned}$$

$$\square$$

### D.2 Proof for Barron Norm of Polynomial

**Lemma 10** ( Lemma 7 restated). *Let*

$$f(x) = \sum_{\alpha, |\alpha| \le P} \left( A_\alpha \prod_{i=1}^d x_i^{\alpha_i} \right)$$

*where $\alpha$ is a multi-index and $x \in \mathbb{R}^d$ and $A_\alpha \in \mathbb{R}$ is a scalar. If $g : \mathbb{R}^d \to \mathbb{R}^d$ is a function such that $g \in \Gamma_W$, then we have $f \circ g \in \Gamma_{PW}$ and the Barron norm can be bounded as,*

$$\|f \circ g\|_{\mathcal{B}(\Omega)} \le d^{P/2} \left( \sum_{\alpha, |\alpha| \le P} |A_\alpha|^2 \right)^{1/2} \|g\|_{\mathcal{B}(\Omega)}^P.$$

*Proof.* Recall from Definition 7 we know that for a vector valued function $g : \mathbb{R}^d \to \mathbb{R}^d$, we have

$$\|g\|_{\mathcal{B}(\Omega)} = \max_{i \in [d]} \|g_i\|_{\mathcal{B}(\Omega)}.$$

Then, using Lemma 6, we have

$$\|f(g)\|_{\mathcal{B}(\Omega)} = \left\| \sum_{\alpha, |\alpha|=0}^P A_\alpha \prod_{i=1}^d g_i^{\alpha_i} \right\|_{\mathcal{B}(\Omega)}$$

$$\le \sum_{\alpha, |\alpha|=0}^P \left\| A_\alpha \prod_{i=1}^d g_i^{\alpha_i} \right\|_{\mathcal{B}(\Omega)}$$

$$\le \sum_{\alpha, |\alpha|=0}^P |A_\alpha| \left\| \prod_{i=1}^d g_i^{\alpha_i} \right\|_{\mathcal{B}(\Omega)}$$

$$\le \sum_{\alpha, |\alpha|=0}^P |A_\alpha| \left\| \prod_{i=1}^d g_i^{\alpha_i} \right\|_{\mathcal{B}(\Omega)}$$

$$\le \sum_{\alpha, |\alpha|=0}^P |A_\alpha| \left( \prod_{i=1}^d \|g_i^{\alpha_i}\|_{\mathcal{B}(\Omega)} \right)$$

$$\le \sum_{\alpha, |\alpha|=0}^P |A_\alpha| \left( \prod_{i=1}^d \|g_i\|_{\mathcal{B}(\Omega)}^{\alpha_i} \right)$$

$$\le \left( \sum_{\alpha, |\alpha|=0}^P |A_\alpha|^2 \right)^{1/2} \left( \sum_{\alpha, |\alpha|=0}^P \left( \prod_{i=1}^d \|g_i\|_{\mathcal{B}(\Omega)}^{\alpha_i} \right)^2 \right)^{1/2} \tag{37}$$

where we have repeatedly used Lemma 6 and Cauchy-Schwartz in the last line. Using the fact that for a multivariate function $g : \mathbb{R}^d \to \mathbb{R}^d$ we have for all $i \in [d]$

$$\|g\|_{\mathcal{B}(\Omega)} \ge \|g_i\|_{\mathcal{B}(\Omega)}.$$

Therefore, from Equation 37 we get,

$$\|f(g)\|_{\mathcal{B}(\Omega)} \le \left( \sum_{\alpha, |\alpha| \le P} |A_\alpha|^2 \right)^{1/2} \left( \sum_{\alpha, |\alpha| \le P} \left( \|g\|_{\mathcal{B}(\Omega)}^{\sum_{i=1}^d \alpha_i} \right)^2 \right)^{1/2}$$

$$\le \left( \sum_{\alpha, |\alpha| \le P} |A_\alpha|^2 \right)^{1/2} \left( \sum_{\alpha, |\alpha| \le P} \left( \|g\|_{\mathcal{B}(\Omega)}^\alpha \right)^2 \right)^{1/2}$$

$$\le d^{P/2} \left( \sum_{\alpha, |\alpha| \le P} |A_\alpha|^2 \right)^{1/2} \|g\|_{\mathcal{B}(\Omega)}^P$$

Since the maximum power of the polynomial can take is $P$ from Corollary 1 we will have $f \circ g \in \Gamma_{PW}$. $\qquad \square$

### D.3 PROOF FOR BARRON NORM ALGEBRA:LEMMA 6

The proof of Lemma 6 is fairly similar to the proof of Lemma 3.3 in Chen et al. (2021)—the change stemming from the difference of the Barron norm being considered

*Proof.* We first show the result for *Addition* and bound $\|h_1 + h_2\|_{\mathcal{B}(\Omega)}$,

$$\|h_1 + h_2\|_{\mathcal{B}(\Omega)} = \inf_{\substack{g_1|_\Omega = h_1, g_1 \in \mathcal{F} \\ g_2|_\Omega = h_2, g_2 \in \mathcal{F}}} \int_{\mathbb{R}^d} (1 + \|\omega\|_2) |\widehat{g_1 + g_2}(\omega)| d\omega$$

$$= \inf_{\substack{g_1|_\Omega = h_1, g_1 \in \mathcal{F} \\ g_2|_\Omega = h_2, g_2 \in \mathcal{F}}} \int_{\mathbb{R}^d} (1 + \|\omega\|_2) |\hat{g}_1(\omega) + \hat{g}_2(\omega)| d\omega$$

$$\leq \inf_{g_1|_\Omega = h_1, g_1 \in \mathcal{F}} \int_{\mathbb{R}^d} (1 + \|\omega\|_2) |\hat{g}_1(\omega)| d\omega + \inf_{g_2|_\Omega = h_2, g_2 \in \mathcal{F}} \int_{\mathbb{R}^d} (1 + \|\omega\|_2) |\hat{g}_2(\omega)| d\omega$$

$$\implies \|h_1 + h_2\|_{\mathcal{B}(\Omega)} \leq \|h_1\|_{\mathcal{B}(\Omega)} + \|h_2\|_{\mathcal{B}(\Omega)}.$$

For *Multiplication*, first note that multiplication of functions is equal to convolution of the functions in the frequency domain, i.e., for functions $g_1 : \mathbb{R}^d \to d$ and $g_2 : \mathbb{R}^d \to d$, we have,

$$\widehat{g_1 \cdot g_2} = \hat{g}_1 * \hat{g}_2 \tag{38}$$

Now, to bound the Barron norm for the multiplication of two functions,

$$\|h_1 \cdot h_2\|_{\mathcal{B}(\Omega)} = \inf_{\substack{g_1|_\Omega = h_1, g_1 \in \mathcal{F} \\ g_2|_\Omega = h_2, g_2 \in \mathcal{F}}} \int_{\mathbb{R}^d} (1 + \|\omega\|_2) |\widehat{g_1 \cdot g_2}(\omega)| d\omega$$

$$= \inf_{\substack{g_1|_\Omega = h_1, g_1 \in \mathcal{F} \\ g_2|_\Omega = h_2, g_2 \in \mathcal{F}}} \int_{\mathbb{R}^d} (1 + \|\omega\|_2) |\hat{g}_1 * \hat{g}_2(\omega)| d\omega$$

$$= \inf_{\substack{g_1|_\Omega = h_1, g_1 \in \mathcal{F} \\ g_2|_\Omega = h_2, g_2 \in \mathcal{F}}} \int_{\omega \in \mathbb{R}^d} \int_{z \in \mathbb{R}^d} (1 + \|\omega\|_2) |\hat{g}_1(z) \hat{g}_2(\omega - z)| \, d\omega dz$$

$$\leq \inf_{\substack{g_1|_\Omega = h_1, g_1 \in \mathcal{F} \\ g_2|_\Omega = h_2, g_2 \in \mathcal{F}}} \int_{\omega \in \mathbb{R}^d} \int_{z \in \mathbb{R}^d} (1 + \|\omega - z\|_2 + \|z\|_2 + \|z\|_2 \|\omega - z\|_2) |\hat{g}_1(z) \hat{g}_2(\omega - z)| \, d\omega dz$$

Where we use $\|\omega\|_2 \leq \|\omega - z\|_2 + \|z\|_2$ and the fact that

$$\int_\omega \int_z \|z\|_2 \|\omega - z\|_2 |\hat{g}_1(z) \hat{g}_2(\omega - z)| d\omega dz > 0.$$

Collecting the relevant terms together we get,

$$\|h_1 \cdot h_2\|_{\mathcal{B}(\Omega)} \leq \inf_{\substack{g_1|_\Omega = h_1, g_1 \in \mathcal{F} \\ g_2|_\Omega = h_2, g_2 \in \mathcal{F}}} \int_{\omega \in \mathbb{R}^d} \int_{z \in \mathbb{R}^d} (1 + \|\omega - z\|_2) \cdot (1 + \|z\|_2) |\hat{g}_1(z)| |\hat{g}_2(\omega - z)| \, d\omega$$

$$= \inf_{\substack{g_1|_\Omega = h_1, g_1 \in \mathcal{F} \\ g_2|_\Omega = h_2, g_2 \in \mathcal{F}}} ((1 + \|\omega\|_2) \hat{g}_1(\omega)) * ((1 + \|\omega\|_2) \hat{g}_2(\omega))$$

Hence using Young's convolution identity from Lemma 11 we have

$$\|h_1 \cdot h_2\|_{\mathcal{B}(\Omega)} \leq \inf_{\substack{g_1|_\Omega = h_1, g_1 \in \mathcal{F} \\ g_2|_\Omega = h_2, g_2 \in \mathcal{F}}} \left( \int_{\omega \in \mathbb{R}^d} (1 + \|w\|_2) \hat{g}_1(\omega) d\omega \right) \left( \int_{\omega \in \mathbb{R}^d} (1 + \|w\|_2) \hat{g}_2(\omega) d\omega \right)$$

$$\implies \|h_1 \cdot h_2\|_{\mathcal{B}(\Omega)} \leq \|h_1\|_{\mathcal{B}(\Omega)} \|h_2\|_{\mathcal{B}(\Omega)}.$$

In order to show the bound for *Derivative*, since $h \in \Gamma_W$, there exists a function $g : \mathbb{R}^d \to \mathbb{R}$ such that,

$$g(x) = \int_{\|\omega\|_\infty \leq W} e^{i\omega^T x} \hat{g}(\omega) d\omega$$

Now taking derivative on both sides we get,

$$\partial_j g(x) = \int_{\|\omega\|_\infty \leq W} i e^{i\omega^T x} \omega_j \hat{g}(\omega) \tag{39}$$

This implies that we can upper bound $|\widehat{\partial_i g}(\omega)|$ as

$$\widehat{\partial_j g}(\omega) = i\omega_j \hat{g}(\omega)$$
$$\implies |\widehat{\partial_j g}(\omega)| \leq W |\hat{g}(\omega)| \tag{40}$$

Hence we can bound the Barron norm of $\partial_j h$ as follows:

$$\|\partial_j h\|_{\mathcal{B}(\Omega)} = \inf_{g|_\Omega = h, g \in \mathcal{F}_W} \int_{\|\omega\|_\infty \leq W} (1 + \|\omega\|_\infty) |\widehat{\partial_j g}(\omega)| d\omega$$

$$\leq \inf_{g|_\Omega = h, g \in \mathcal{F}_W} \int_{\|\omega\|_\infty \leq W} (1 + \|\omega\|_\infty) |W \hat{g}(\omega)| d\omega$$

$$\leq W \inf_{g|_\Omega = h, g \in \mathcal{F}_W} \int_{\|\omega\|_\infty \leq W} (1 + \|\omega\|_\infty) |\hat{g}(\omega)| d\omega$$

$$\leq W \|h\|_{\mathcal{B}(\Omega)}$$

In order to show the preconditioning, note that for a function $g : \mathbb{R}^d \to \mathbb{R}$, if $f = (I - \Delta)^{-1} g$ then we have then we have $(I - \Delta)f = g$. Using the result form Lemma 12 we have

$$(1 + \|\omega\|_2^2)\hat{f}(\omega) = \hat{g}(\omega) \implies \hat{f}(\omega) = \frac{\hat{g}(\omega)}{1 + \|\omega\|_2^2}.$$

Bounding $\|(I - \Delta)^{-1} h\|_{\mathcal{B}(\Omega)}$,

$$\|(I - \Delta)^{-1} h\|_{\mathcal{B}(\Omega)} = \inf_{g|_\Omega = h, g \in \mathcal{F}} \int_{\omega \in \mathbb{R}^d} \frac{1 + \|\omega\|_2}{(1 + \|\omega\|_2^2)} \hat{g}(\omega) d\omega$$

$$\leq \inf_{g|_\Omega = h, g \in \mathcal{F}} \int_{\omega \in \mathbb{R}^d} (1 + \|\omega\|_2) \hat{g}(\omega) d\omega$$

$$\implies \|(I - \Delta)^{-1} h\|_{\mathcal{B}(\Omega)} \leq \|h\|_{\mathcal{B}(\Omega)}. \qquad \square$$

**Corollary 1.** *Let $g : \mathbb{R}^d \to \mathbb{R}$ then for any $k \in \mathbb{N}$ we have $\|g^k\|_{\mathcal{B}(\Omega)} \leq \|g\|_{\mathcal{B}(\Omega)}^k$. Furthermore, if the function $g \in \mathcal{F}_W$ then the function $g^k \in \Gamma_{kW}$.*

*Proof.* The result from $\|g^k\|_{\mathcal{B}(\Omega)}$ follows from the multiplication result in Lemma 6 and we can show this by induction. For $n = 2$, we have from Lemma 6 we have,

$$\|g^2\|_{\mathcal{B}(\Omega)} \leq \|g\|_{\mathcal{B}(\Omega)}^2 \tag{41}$$

Assuming that we have for all $n$ till $k - 1$ we have

$$\|g^n\|_{\mathcal{B}(\Omega)} \leq \|g\|_{\mathcal{B}(\Omega)}^n \tag{42}$$

for $n = k$ we get,

$$\|g^k\|_{\mathcal{B}(\Omega)} = \|gg^{k-1}\|_{\mathcal{B}(\Omega)} \leq \|g\|_{\mathcal{B}(\Omega)} \|g^{k-1}\|_{\mathcal{B}(\Omega)} \leq \|g\|_{\mathcal{B}(\Omega)}^k. \tag{43}$$

To show that for any $k$ the function $g^k \in \Gamma_{kW}$, we write $g^k$ in the Fourier basis. We have:

$$g^k(x) = \prod_{j=1}^{k} \left( \int_{\|\omega_j\|_\infty \leq W} \hat{g}(\omega_j) e^{i\omega_j^T x} d\omega_j \right)$$

$$= \int_{\|\omega\|_\infty \leq kW} \left( \int_{\sum_{l=1}^{k} \omega_l = \omega} \Pi_{j=1}^{k} \hat{g}(\omega_j) d\omega_1 \ldots d\omega_k \right) e^{i\omega^T k} d\omega$$

In particular, the coefficients with $\|\omega\|_\infty > kW$ vanish, as we needed. $\qquad\square$

**Lemma 11** (Young's convolution identity). *For functions $g \in L^p(\mathbb{R}^d)$ and $h \in L^q(\mathbb{R}^d)$ and*

$$\frac{1}{p} + \frac{1}{q} = \frac{1}{r} + 1$$

*where $1 \leq p, q, r \leq \infty$ we have*

$$\|f * g\|_r \leq \|g\|_p \|h\|_q.$$

*Here $*$ denotes the convolution operator.*

**Lemma 12.** *For a differentiable function $f : \mathbb{R}^d \to \mathbb{R}$, such that $f \in L^1(\mathbb{R}^d)$ we have*

$$\widehat{\nabla f}(\omega) = i\omega \hat{f}(\omega)$$

