# OpenReview forum: "Neural Network Approximations of PDEs Beyond Linearity: Representational Perspective"
_ICLR.cc/2023/Conference — Submitted to ICLR 2023_

### Official Review · Reviewer_AUeL · 2022-10-18

**Confidence:** 3
**Correctness:** 3
**Technical Novelty And Significance:** 2
**Empirical Novelty And Significance:** Not applicable
**Recommendation:** 3

**Clarity, Quality, Novelty And Reproducibility:**

Overall, the manuscript provides a clear overview of the proof techniques used for the main theorem. Unfortunately, the paper is currently not very well written and exhibits a lot of spelling and grammatical mistakes, which make reading less smooth and can lead to some confusion. Examples for this are in Theorem 4, where it is not clear, whether the constant $R$ can be chosen or is only guaranteed to exist and in Theorem 1, whether the complexity bound holds for $\epsilon\to0$ or $d\to\infty$.

The proof techniques are very similar to previous works, where the difference is to go from a quadratic energy to an energy, which is strongly convex with upper bounded second derivative, i.e., admits upper and lower bounds of the squared norm. Hence, the techniques appear to be not novel, that being said, this is a normal way to go, when generalizing result for linear to non-linear problems. Thus, I don’t think this is a reason to reject this paper.

**Strength And Weaknesses:**

The paper addresses the timely and important question of the approximation properties of neural network related function classes of solutions of variational problems. I believe that the extension to general variational problems is a valuable contribution. Main weaknesses of the manuscript include:
* The work follows existing works on linear problems very closely.
* The consequences of the main theorem are not well elaborated. For example, no theorem bounding the Barron norm or the complexity of a shallow network required for eps approximation is given. Although these steps seem straight forward, they would be important to be carried out, especially in light of Theorem 4 the error does not decay to zero as discussed by the authors.
* The paper is currently not very well written and exhibits a lot of spelling and grammatical mistakes, which and should be addressed.

**Summary Of The Paper:**

The manuscript studies the approximation properties of solutions of variational problems with Barron functions. In particular, this includes solutions of nonlinear PDEs like the p Laplace equation. It follows recent works on approximation properties for solutions of linear PDEs and extends this analysis to general variational problems. More precisely, it considers a preconditioned gradient descent of the variational energy in the Sobolev space $H^1_0$, which converges linearly, and bounds the Barron norm of the iterates, which relies on an assumption on the variational energy. Overall, this provides an estimate on the Barron norm required for the approximation of the solution of the variational problem up to a given accuracy. By standard arguments, this also implies upper bounds on the complexity of shallow networks with sigmoidal or ReLU activation for these target class and provides way of analysing the dependence of the convergence properties in dependence of the dimension of the problem.

**Summary Of The Review:**

The paper addresses the timely and important question of the approximation properties of neural network related function classes of solutions of variational problems. I believe that the extension to general variational problems is a valuable contribution. That being said, I believe that the manuscript in its form is not ready for publication. My main concerns are the following:

* *Implications of results:* The actual implications are not well developed. In particular, it would be important to elaborate the consequences of Theorem 4. In particular, it is not very clear, whether Theorem is able to establish estimates for $\varepsilon\to0$. Interesting implications include a precise formulation of the Barron norm required for $\varepsilon$ approximation of a solution of a variational problem and complexity bounds for neural networks for $\varepsilon$ approximation. This would could greatly improve the insights of the results and would make the results more applicable for future works. I know that to some extend this is done in Remarks 1 and 2, but it is not clear to me, whether this holds for arbitrary $\epsilon$ and if so how to proof this.
* *Inhomogeneous problems:* Can you also treat inhomogeneous variational problems? This seems to be crucial for the following reason: Consider the Laplace equation, then solution of the homogeneous problem are harmonic hence smooth and can be approximated at a dimension independent rate, where the constants might depend on the dimension (follows from standard arguments like Yarotsky).
* *Clarity and quality of writing:* Unfortunately, the paper is currently not very well written and exhibits a number of spelling and grammatical mistakes as well as unintuitive formulations, which make reading less smooth and can lead to some confusion. I attached a list of specific observations below.

Specific questions:
* Can you provide a theorem that bounds the Barron norm and complexity of a network required for $\epsilon$ approximation?
* Why do you state your results in the abstract and introduction for $L^2$ and not for $H^1$ approximation?
* In Assumption 1: Is this supposed to hold for some $\epsilon_L<\lambda$? Then I would formulate it that the assumption is that $\epsilon_L := \sup \dots < \lambda$.
* In Theorem 4: it is not clear to me what „considering a constant $R \le \dots$ “ means. Can this be chosen? If so, can it be set to 0? If it can not be chosen, then I would maybe say „$R = \dots$“.
* Can you treat non-homogeneous problems?
* What is the benefit of working with $\tilde L$ and not with $L$?

Specific comments:
* The formulation „neurally simulating (preconditioned) gradient descent“ is confusing to me. Although, it sounds very fancy, I do not understand what it means, since form my understanding you use this for preconditioned gradient descent in a Sobolev space without having a direct connection to neural networks nor to simulation of something. When using such a terminology it should be better explained.
* In the introduction, Sirignano&Spiliopoulos (2018) is given as a reference for a work minimizing the variational formulation of a PDE. However, they minimize the residual energy, which is fundamentally different to the variational energy.
* It would make sense in Lemma 1 to say that (3) is to be understood in the weak sense.
* In Theorem 4: I wonder whether $\tilde\epsilon$ is a good symbol, since epsilons are usually going to zero, which this quantity is not.
* You claim to „subsume and substantially generalize“ prior works. In general, I would be very careful with judging the impact of any work, but this claim seems not entirely correct. From what I can see you extend their analysis to general variational problems, but only study homogeneous equations. Hence, prior results on linear but inhomogeneous problems are not special cases of your results.
* Further references of interest: E&Wojtowytsch (2022), Grohs&Herrmann (2022)
* Regarding Yarotsky: Yarotsky provides an approximation rate of $O(N^{-r/d})$, where $r$ is the regularity of the target function; if the solutions of the PDEs are smooth, which is for example the case for harmonic functions, then approximation with a power law with exponent independent of the dimension is possible; hence I disagree with your statement that the networks must by exponentially large; further, I do not know what you mean by Yarotsky's results being non-parametric
* It could be a consideration to include the appendix in the normal PDF rather than the supplement.

Throughts regarding formatting and writing:
* In the list of references, the titles should be case sensitive, for example „PDEs“ instead of „pdes“.
* Try to avoid double brackets.
* On page 1: it is stated that "... is at least as expensive as the neural network required to represent it.". I think this is not very clearly written, maybe it would be better to say "In general the complexity of fitting a NN grows with its size"?
* Section 2:
    * Should be $\Omega\subset\mathbb R^d$ and $\partial\Omega$
    * Generally better to use $u\colon\Omega\to\mathbb R$ instead of $u:\Omega\to\mathbb R$ ("\colon" instead of ":")
    * Below Definition 1 it would be nice to add in what space the problem is well posed
    * In Lemma 1 it should be "the minimizer $u^\star$
* Section 3:
    * In the first sentence it should b "... been a growing line of works that utilize neural"
    * Regarding "This is a great and promising direction..." I would in general shy away from judging the quality of other research in a research paper
    * It should be "equation and Lu & Lu (2021) extend their analysis"
    * It should be "Schrödinger" (\ "{o})
    * When using a bibtex style that gives the authors names, I would not write "in [authors names]" but simply "[authors names]" (e.g., "In Marwah et al. (2021)")
* Section 4:
    * I would try to not start a sentence with a symbol (in your case $C^\infty$)
    * In Definition 3 it should be $g\in L^2$ and I would rather write $\nabla g\in L^2$
    * In Theorem 3 I would introduce $\sigma$ before the statement.
* Section 6:
    * "neurally unfolding": As said before, I do not understand what is meant with this; I think the description as preconditioned GD in a Sobolev space is perfectly clear.
    * Lemma 4: there should be a full stop after "Definition 3"
    * The reference to Equation 28: is this where it should refer to? If so, referencing to something burried in the appendix for a high level idea is not ideal.

---

> ### Author Response · Authors · 2022-11-09
> **Thank you for your review! (Part 1)**
>
> Thank you for the detailed comments and concrete suggestions for improving the writing. Your suggestions have helped us to improve the clarity of the exposition (in the revised draft, edits are shown in blue).
>
> Below we hope to clarify what we believe may be the primary points of confusion and thus to resolve your concerns. Please let us know if anything remains unclear.
>
> **The work follows existing works on linear problems very closely**:
>
> While the general proof idea of simulating some kind of gradient-descent-like process in an appropriate Hilbert space builds on ideas from prior works, adapting these techniques to our setting required developing new  technical machinery. Examples include identifying the right proxies of convexity of the variational loss (Lemma 2); and choosing the right preconditioner to get progress in the “right norm” (see Lemma 3, and the comments before it).
>
> **“Implication of results”, $\epsilon \to 0$, “what is the benefit of working with $\tilde{L}$ and not $L$”, “I wonder whether $\tilde{\epsilon}$ is a good symbol”**:
>
> There seems to be some confusion about whether $\epsilon$ can be assumed to go to 0, and what implications it has for the main theorem. Below we will clarify a few points, which we hope may narrow down the possible sources of confusion.
>
> * In both Theorem 1 and Theorem 4, $\epsilon$ can be arbitrarily small—we made this point more explicit in the updated draft in both theorems, as well as edited Theorem 4 slightly to state the dependence on $\epsilon$ more explicitly. (Note this is a simple rewrite, as $\epsilon$ and $T$ are related by the relation stated in the Theorem: $\epsilon := \frac{1}{\lambda}\left(1 - \frac{\lambda^5}{(1 + C_p) \Lambda^4}\right)^T \mathcal{E}(u_0)$. Note also this is merely a presentational preference: the original way we wrote it and the way we edited it are completely equivalent.)
>
> * If $\tilde{\epsilon} = 0$, this immediately gives a Barron norm upper bound for an $\epsilon$-approximation to the solution of the PDE, for any small enough $\epsilon$. As in Remark 2, $\tilde{\epsilon}$ is a term that cannot be driven to 0: This term arises because the function $L$ may not **itself** have the property that composing with it a function $g$ does not result in a function with much bigger Barron norm than $g$. However, in Assumption 1 we allow that there is some function $\tilde{L}$ close to $L$ (in the sense that $\sup_{x \in \mathbb{R}^d} ||\nabla L(x) - \nabla \tilde{L}(x)||_2 \leq \epsilon_L||x||_2$), s.t. composing with **it** does not result in a function with a much bigger Barron norm than $g$. This is quite natural—e.g. Lemma 7 shows that composing polynomials of bounded degree with functions $g$ results in a Barron norm that is not much larger than that of $g$—thus Assumption 1 allows us to deal with equations for which $L$ is not **exactly** a polynomial, only **close to** one. We think of $\tilde{\epsilon}$ as something small, which is the reason for the choice of variable name. A completely analogous assumption (with the same variable name!) was employed in prior work for linear PDEs, e.g. Marwah et al, where it is assumed the coefficients are only approximately small neural networks. This is also the reason we need to work with $\tilde{L}$ instead of $L$: our proof involves tracking the increase in Barron norm after each step of preconditioned gradient descent: this is only possible if we are performing gradient descent on the equation defined through $\tilde{L}$, **not** the one defined through $L$.
>
> **“Inhomogeneous problems”**:
>
> The usual definition of a homogeneous PDE is one for which if $u:\Omega \to \mathbb{R}$ is a solution, so is $c \cdot u$ for any non-zero constant $c$. The PDE in our case certainly does not have to satisfy this property: consider e.g. the convex function $L(x) = x_1^4 + x_2^2$. A short calculation shows
> $$\mbox{div}(\nabla L(\nabla u(x))) = 12 (\partial_1 u(x))^2  \partial_{11} u(x) + 2 \partial_{22} u(x).$$
> If we consider $c \cdot u$, the first term scales by $c^3$, and the second by $c$, so the resulting equation is not homogeneous.
>
> Please let us know if this addresses your question or if you have any lingering concerns.

---

> ### Author Response · Authors · 2022-11-09
> **Thank you for your review (Part 2)**
>
> **“Regarding Yarotsky, nonparametric vs parametric rates”**:
>
> To be clear, we don’t only care about rates (i.e., the dependence on $\epsilon$). If we only cared about rates, then exponential dependence on dimension in the constants would be ok. For example, you wrote: “...solutions of the homogeneous problem are harmonic hence smooth and can be approximated at a dimension independent rate, where the constants might depend on the dimension (follows from standard arguments like Yarotsky).” However, the primary focus of this work is to **avoid the curse of dimensionality** (see exposition in the abstract and introduction)—thus exponential dependence on dimension in the constants is **not allowed**. Whenever we use O(.) notation, we are careful to specify what we suppress, e.g. in Remark 3: “...if we think of $p, \Lambda, \lambda, C_p, k, ||u_0||_{B(\Omega)}$ as constants…”—but the dependence on dimension is **never** suppressed.
> Non-parametric rates will have an exponential dependence on dimension—either in the **rate**, or in the **constant** in front of the rate—where the constant in the exponent improves depending on smoothness properties of the function.  (E.g., as the reviewer said, $C n^{-r/d}$). The terms *"curse of dimensionality"* are used both in Marwah et al and Chen et al, and we’ve updated the draft to also use this nomenclature and avoid the term “non-parametric”—hopefully this is clearer.
>
> **“Neurally unfolding/simulating”**:
>
> This is a non-technical expression, borrowed from  the deep learning literature, see e.g. Hershey, Le Roux, Weninger: “Deep unfolding: Model-based inspiration of novel deep architectures” (2014). To signal the informality of the term, it is commonly placed  in quotations (a convention that we adopt here). We unpack this term immediately upon its usage in both the abstract and intro (“...namely, we show that each of the iterates can be represented by a neural network with Barron norm not much worse than the Barron norm of the previous iterate—along with showing a bound on the number of required steps…”). The term is intended to suggest that we simulate (i.e. represent) the iterates of preconditioned gradient descent using neural networks, and bound the size of each iterate (by the relationship between Barron norm and size of a neural network)..
>
> **“Why do you state your results in the abstract and introduction for $L^2$ and not for $H^1$ approximation?”**:
>
> The $L^2$ approximation is weaker than $H^1$ approximation, since $||u||_2 \leq C_p ||\nabla u||_2$, and the $H^1$ norm is not widely known in the machine learning community. (Thus, having it in the abstract/introduction before defining it would likely catch off-guard a portion of the readers.)
>
> **“In Theorem 4: it is not clear to me what „considering a constant  $R \leq $…“ means. Can this be chosen? If so, can it be set to 0? If it can not be chosen, then I would maybe say “$R=$…”**:
>
> You are right, we should have said $R = …$. In fact, the theorem seems cleaner to read without even introducing $R$—we’ve edited the draft to that effect.
>
> **More minor writing/formatting suggestions:**
>
> Most of these seemed reasonable to us and we updated the draft accordingly. We’ve also attached the appendix as part of the main paper for easier access to the appendix.

---

> ### Author Response · Authors · 2022-11-18
> **Can we help address some other concerns?**
>
> We thank you again for your detailed review and comments! We hope that our reply has addressed your concerns regarding our paper. Please let us know if there are still questions remaining that we can resolve.
>
> Additionally, we have incorporated most of your writing suggestions to improve the clarity of our manuscript and hope that it addresses your concerns regarding the quality of the exposition. Since we are nearing the end of the paper edit phase of the review cycle (11/18), please let us know if the changes are appropriate and if there is anything we can do to improve the manuscript.

---

> ### Comment · Reviewer_AUeL · 2022-11-29
> **Reply to authors response**
>
> Dear authors,
>
> thanks for your reply and the adjustments of the manuscript, which have improved the overall readability.
>
> Regarding my point of inhomogeneous problems, I was thinking about non zero right hand sides of the PDE (if formulated like in (3)). I think it is interesting to develop the theory further for equations of the form $-\operatorname{div}(\nabla L(\nabla u)) = f$, but would never suggest a rejection because of this.
>
> However, my main criticism in the initial review about the implications of the results remains valid. Most importantly, I am still not convinced that the statements in Theorem 1 and Remark 3 hold for all $\epsilon>0$ (small enough). At least I can not deduce them from Theorem 4, since the overall error $\epsilon+\tilde\epsilon$ does not decrease to $0$ in Theorem 4.
>
> To elaborate my concern further, recall that in Remark 3, it is stated, that by Theorem 4 the solution $u^\star$ of the PDE can be $\epsilon$ approximated in $L^2$ by a function with Barron norm $O((dB_L)^{p^{\log(1/\epsilon)}})$. However, Theorem 4 ensures the existence of a function $u_\epsilon$ (which is given by $u_T$ in the notation of Theorem 4) with Barron norm $O((dB_L)^{p^{\log(1/\epsilon)}})$ such that $\lVert u_\epsilon - u^\star \rVert_{L^2(\Omega)} = O(\epsilon + \tilde\epsilon)$. Here, $\tilde \epsilon$ depends on $\epsilon$ and Theorem 4 establishes the estimate $\tilde\epsilon = O(\eta^{\log(1/\epsilon)})$, where $\eta>1$, which diverges to $+\infty$ for $\epsilon\to0$. Since $\epsilon+\tilde\epsilon$ can not shown to be arbitrarily small, I do not see how Theorem 4 can be used to establish statements about arbitrarily good approximations. Note that Remark 3 is the proof Theorem 1, which is advertised as the main result and would probably be the result people would like to use in future work. Therefore, I believe that it requires a solid proof, which is currently not given within the manuscript. That being said, I do not rule out the possibility that there is an argument that establishes the claims in Remark 3 and Theorem 1.
>
> Additionally, I want to bring up one point, that I only encountered when going over the revised version: The complexity estimates established in the manuscript do not depend exponentially on the dimension $d$ of the problem, but they appear to depend exponentially on the accuracy $\epsilon$. Note that this is in difference to the results Marwah et. al., where the complexity is $O(d^{\log(1/\epsilon)})$ compared to $O((dB_L)^{p^{\log(1/\epsilon)}})$ in this work, which grows exponentially for $\epsilon\to0$. I did not realize this in my initial review, but this significantly weakens the claim of the paper, which addresses the curse of dimensionality in the neural network approximation of solutions of PDEs.
>
> After all, I can still not recommend acceptance of the manuscript in its current form. The main reasons for this are the insufficient proof of Theorem 1, where the validity is currently unclear to me. Further, also with a valid proof of Theorem 1 the implications and the significance of this result – also in the light of the exponential dependence in $\epsilon$ – are not entirely clear and would deserve further elaboration. Nevertheless, I like the line of work deducing approximation results by studying variational energies in function space and would like the manuscript to see published after sufficient improvements.
>
> With kind regards

---

> > ### Author Response · Authors · 2022-11-29
> > **Thank you for your reply!**
> >
> > Thank you for your response. We are happy to know that you found our adjustments to the paper acceptable and think that it improves the overall readability.
> >
> > Please find below the answers to your other concerns:
> >
> > **Re: Theorem 1 and Remark 3 to hold for small enough $\epsilon > 0$.**
> > You are correct that $\epsilon$ cannot be driven to zero **unless** $\tilde{\epsilon}$ is zero. We believe we were forthright about this both in the rebuttal and the draft (e.g. we say in Remark 2: “The error term $\tilde{\epsilon}$ stems from the approximation that we make between $\tilde{L}$ and $L$, which grows as $T$ increases..”). If you feel like there’s a particular choice of words that you believe would make this clearer, we are happy to add something to that effect.
> > We note that this type of guarantee is **not unusual**: $\tilde{\epsilon} = 0$ corresponds to assuming the function $L$ *itself* has the property that composing with it a function $g$ does not result in a function with much bigger Barron norm than $g$—for example if $L$ is a multivariate polynomial (Lemma 7). The special case of $\tilde{\epsilon}=0$ in the case of linear PDEs is assuming the coefficients of the linear PDE have small Barron norm (as opposed to being *merely close* to a function with a small Barron norm)—so this kind of statement just allows for a little ”assumption mismatch”. In fact, the prior work Marwah et al has an **identical** term $\tilde{\epsilon}$ that does not go to 0, and gets worse as $\epsilon$ gets smaller.
> > Another note is that Theorem 1 is an **informal theorem statement**, and is also labeled that way in the paper. The main result of the paper is the formal version of that theorem: Theorem 4 — again, it is explicitly **labeled as “(Main result)”**. This organization reflects an intentional expository choice on our part, made to quickly and informally convey the key result of our paper, without introducing a lot of notation and technical nuance. This is common practice in theoretically-heavy machine learning papers.
> >
> >
> > **Re: Worse rates than Marwah et al.**
> > You are correct, the dependence on $\epsilon$ is worse compared to Marwah et al for $p \neq 1$ (who consider the case of linear PDEs in their paper). However, both results can **only** be considered to evade the curse of dimensionality if $\epsilon$ (the target approximation accuracy) is thought of as a constant (i.e. independent of $d$)—or growing extremely slowly with $d$. We are happy to add a sentence/remark to this effect in the paper. (Improving the dependence on $\epsilon$ in the exponent is fertile ground for further work.)
> >
> > **Re: Inhomogeneous case**
> > Thank you for your clarification, we agree that it is interesting to develop our theory further.

---

> > > ### Comment · Reviewer_AUeL · 2022-11-30
> > > **Response**
> > >
> > > Dear authors,
> > >
> > > thanks again for your response, which resolved parts of my concern. I think I was not aware that Remark 3 was making the assumption that $\epsilon_L=0$ and hence $\tilde\epsilon=0$. Just reading Theorem 4 and Remark 3 this was not clear and I could therefore not follow this argument. I would you advise to point out the assumption $\epsilon_L=0$ in Remark 3 as this can easily prevent a mathematical compilation error when reading your manuscript.
> > >
> > > Regarding the exponential dependence on $\epsilon$: it might be good to add an explanation, what the benefit of a result without the curse of dimension in the constants but rather in $\epsilon$ is compared to a classical result, where the approximation rate in $\epsilon$ would be given by a power law decay and the constants would grow in the dimension are.
> > >
> > > Reading your work once more, there is one more short question, that arose, which I am sure you can answer: What does prevent me from choosing $u_0=0$ in Theorem 4? Clearly, it can not be allowed, since otherwise the bound on the Barron norm would be constant $0$ and hence $u_T=0$, since the Barron norm only vanishes for the zero function. I hope you can help me out on this once more, in which case I would be willing to raise my score.
> > >
> > > With best regards

---

> > > > ### Author Response · Authors · 2022-12-02
> > > > **Thank you for your response!**
> > > >
> > > > Thank you for your reply. We greatly appreciate all the feedback so far, and we are glad our clarifications are helping! We also apologize for the somewhat belated response: we are all at NeurIPS so our responses are a bit delayed.
> > > >
> > > > First, we are happy to update remark 3 to say that $\tilde{\epsilon} = 0$. More specifically we will update Remark 3 to say the following: “As in the informal theorem, if we think of $p, \Lambda, \lambda, C_p, k, \|u_0\|_{\mathcal{B}(\Omega)}$ to be constants and $\tilde{\epsilon} = 0$, then the theorem implies that $u^\star$ can be $\epsilon$-approximated in the $L^2(\Omega)$ sense…”.
> > > >
> > > > Regarding the exponential dependence on $\epsilon$: yes, we are also happy to add a note on the improved dependence on $d$ in the constant compared to nonparametric rates, but the worse exponent in $\epsilon$.
> > > >
> > > > Finally, regarding taking $u_0 = 0$ in Theorem 4: thank you for this question—we actually realized that there is a minor oversight in Assumption 1 due to your question. There is technically nothing preventing us to set $u_0$ in Theorem 4: however, Assumption 1 as stated would force $u_0=0$ to be a solution to the PDEs we consider, so no “iterations” would be needed. (For example, $L$ being a polynomial with a constant term would not satisfy Assumption 1.)
> > > > Fortunately, it’s very easy to modify Assumption 1 to allow equations for which $u_0=0$ is not a solution by adding a constant term in the growth bound as follows:
> > > >
> > > > **Assumption 1 (updated)**:     The function $L$ in Definition 1 can be approximated by a function
> > > >     $\tilde{L}:\mathbb{R}^d \to \mathbb{R}$ such that there exists a constant $\epsilon_L \in [0, \lambda)$ with $\sup_{x \in \mathbb{R}^d} \\|\nabla L(x) - \nabla \tilde{L}(x)\\|_2 \leq \epsilon_L \\|x\\|_2$.
> > > >
> > > > Furthermore, we assume that $\tilde{L}$ is such that for any $g \in H_0^1(\Omega)$, we have $\tilde{L} \circ g \in H_0^1(\Omega)$, $\tilde{L} \circ g \in \mathcal{F}$ and
> > > > $$
> > > > \\|\tilde{L} \circ g\\|_{\mathcal{B}(\Omega)} \leq B\_{\tilde{L}}\\|g\\|\_{\mathcal{B}(\Omega)}^p + C\_{\tilde{L}}
> > > > $$
> > > > for some constants $B\_{\tilde{L}}, C\_{\tilde{L}} \geq 0$, and $p \geq 1$. Furthermore, if $g \in \Gamma\_{W}$ then $\tilde{L} \circ g \in \Gamma\_{kW}$ for a $k >0$.
> > > >
> > > > Note that, in the case of $L$ being a polynomial, $C_{\tilde{L}}$ will be the constant term of the polynomial.
> > > >
> > > > We note that propagating this change to Assumption 1 is very straightforward, in particular it primarily affects the results of Lemma 5:
> > > >
> > > > **Lemma 5 (updated)**. Consider the updates in Equation 8, if $\tilde{u}\_t \in \Gamma\_{W\_t}$
> > > >     then for all $\eta \in (0, \frac{\lambda^3}{(C_p + 1) \Lambda^3}]$ we have $\tilde{u}\_{t+1} \in \Gamma\_{kW\_t}$ and the Barron norm of $\tilde{u}\_{t+1}$ can be bounded as
> > > > $$
> > > > \\|\tilde{u}\_{t+1}\\|_{\mathcal{B}(\Omega)} \leq \left(1 + \eta d (kW\_t)^2 B\_{\tilde{L}}\right)
> > > > \left(\\|\tilde{u}\_t\\|\_{\mathcal{B}(\Omega)}+ C\_{\tilde{L}}\right).
> > > > $$
> > > >
> > > > Given the updated result, and expanding upon the recursion the Barron norm for $u_T$ can be bounded by
> > > > $$
> > > >     \\|u_{T}\\|\_{\mathcal{B}(\Omega)} \leq
> > > >                     \left(1 + \frac{\lambda^3}{(C_p + 1)\Lambda^3} dk^{2}W\_0^2 B\_{\tilde{L}}\right)^{p^T+1} \left(\\|u\_0\\|\_{\mathcal{B}(\Omega)}^{p^T} + C\_{\tilde{L}}^{p^{T}}\right)
> > > > $$
> > > >
> > > > With the updated Assumption and if $u_0 = 0$, the Barron norm of $u_T$ will not be equal to zero. We thank the reviewer again for bringing this point up and making our paper stronger.

---

> > > > > ### Comment · Reviewer_AUeL · 2022-12-07
> > > > > **Follw up question**
> > > > >
> > > > > Dear authors,
> > > > >
> > > > > thanks a lot for you reply. I will take more time to digest your answer, but have a very short question regarding your now: Could you please elaborate in more details, why choosing $u_0=0$ implies that $0$ is a solution of the PDE?
> > > > >
> > > > > Thanks again a lot for your elaborations, best regards

---

> > > > > > ### Author Response · Authors · 2022-12-07
> > > > > > **Reply to Follow up**
> > > > > >
> > > > > > Certainly, to be clear, we claim that if Assumption 1 is as currently stated in the paper and $\epsilon_{L} = 0$, that is the function $L$ is such that for all functions $g \in H_0^1(\Omega)$ we have $\\|L \circ g\\|\_{\mathcal{B}(\Omega)} \leq B_{L}^p \\|g\\|\_{\mathcal{B}(\Omega)}$, then $u = 0$ is a solution to the PDE in Equation (3). This follows, since the above assumptions imply that $\nabla L(0) = 0$, as we show below:
> > > > > >
> > > > > > **Lemma**: If for any $g \in H\_0^1(\Omega)$ such that $g \in \Gamma\_{W}$ for $W$ finite, we have $\\|L \circ g\\|\_{\mathcal{B}(\Omega)} \leq B\_{L} \\|g\\|^p\_{\mathcal{B}(\Omega)}$, for some constants $B\_{L} \geq 0$ and $p \geq 1$, then we have that $\nabla L(0) = 0$.
> > > > > > **Proof:** Note that we have $\\|L \circ g\\|\_{\mathcal{B}(\Omega)} \leq B\_{L} \\|g\\|^p\_{\mathcal{B}(\Omega)}$. Therefore, using the “Derivative” bullet from Lemma 6, we also $\\|\nabla L \circ g\\|\_{\mathcal{B}(\Omega)} \leq W B_{L} \\|g\\|^p\_{\mathcal{B}(\Omega)}$.
> > > > > > Hence, if $g(x) = 0$ for all $x \in \Omega$, it would imply that $\\|\nabla L \circ g\\|\_{\mathcal{B}(\Omega)} \leq 0$ as well, and since Barron norm is always nonnegative, we have $\\|\nabla L(g)\\|\_{\mathcal{B}(\Omega)} = 0$. Thus, we have the claim of the lemma that $\nabla L(0) = 0$.
> > > > > >
> > > > > >
> > > > > >
> > > > > > Note that from the above lemma it’s immediate that $u=0$ is a solution to (3), which has the form
> > > > > > $$-\text{div}(\nabla L (\nabla u)) = 0.$$
> > > > > >
> > > > > > With the modified Assumption 1 in the previous reply, $g(x) = 0$ for all $x \in \Omega$ no longer implies that $\nabla L(0) = 0$, thus $u_0 = 0$ is no longer necessarily a solution.
> > > > > >
> > > > > > We hope that this clarification helps!

---

> > > > > > > ### Comment · Reviewer_AUeL · 2022-12-11
> > > > > > > **Once again a question**
> > > > > > >
> > > > > > > Dear authors,
> > > > > > >
> > > > > > > thank you once more for your answer and sorry for my late reply, I was travelling for a conference these days and am on my way back now. I am sorry for once again asking a follow up question, I am just trying to ensure the presented statements are correct. In the above Lemma I wonder, what happens for the special case $d=1$ and $L=\operatorname{id}$. This choice should be covered by the Lemma with the choices $B_L=1$ and $p=1$ as I believe, however, $\nabla L(0) = 1 \ne0$. More generally, by Lemma 7 any polynomial $L$ should satisfy Assumption 1 and hence fall into the setting of the Lemma above, but surely not for any polynomial it holds that $\nabla L(0)=0$. Am I overlooking something here?
> > > > > > >
> > > > > > > One possible source of error could be that Lemma 6 guarantees $\lVert \nabla (L\circ g) \rVert_{\mathcal B(\Omega)} \le W B_L \lVert g \rVert_{\mathcal B(\Omega)}$. Note that $\nabla (L\circ g)(x) = Dg(x)^\top\nabla L(g(x))$. Hence, I do currently not see how this statement holds for $(\nabla L)\circ g \ne \nabla (L\circ g)$.
> > > > > > >
> > > > > > > Again, I might be missing an important point in your arguments and am sorry for asking for further elaborations once more, but can not raise my score based on this new Lemma at the moment.
> > > > > > >
> > > > > > > With kind regards

---

> > > > > > > > ### Author Response · Authors · 2022-12-11
> > > > > > > > **Reply to question**
> > > > > > > >
> > > > > > > > Of course!
> > > > > > > >
> > > > > > > > Perhaps the more general point is this: *"More generally, by Lemma 7 any polynomial $L$ should satisfy Assumption 1 and hence fall into the setting of the Lemma above, but surely not for any polynomial it holds that $\nabla L(0)=0$"*.
> > > > > > > >
> > > > > > > > You are correct! We mentioned this in our reply "Thank you for your response!" that "a polynomial with a constant term would not satisfy Assumption 1." However, the updated Assumption 1 allows a constant that takes care of this.
> > > > > > > >
> > > > > > > > Note that Lemma 7 will also get slightly updated to match the notation in the updated Assumption 1. The Lemma 7 statement will be updated to the following,
> > > > > > > >
> > > > > > > >
> > > > > > > > **Lemma 7 (updated)**: Let $f(x) = \sum\_{\alpha, |\alpha|\leq P} \left(A\_{\alpha}\prod_{i=1}^d x_i^{\alpha_i}\right)$ where $\alpha$ is a multi-index and $x \in \mathbb{R}^d$. If $g: \mathbb{R}^d \to \mathbb{R}^d$ is such that $g \in \Gamma_W$, then we have $f \circ g \in \Gamma_{PW}$ and the Barron norm can be bounded as,
> > > > > > > > $$
> > > > > > > >  \\|f \circ g\\|\_{\mathcal{B}(\Omega)} \leq d^{P/2}\left(\\sum\_{\alpha, |\alpha| = 1}^P |A\_{\alpha}|^2 \right)^{1/2}\\|g\\|\_{\mathcal{B}(\Omega)}^P + |A\_0|.
> > > > > > > > $$
> > > > > > > >
> > > > > > > >
> > > > > > > > Here the main change to the proof is in Equation (37) which now will be,
> > > > > > > >  $$ \\|f \circ g\\|\_{\mathcal{B}(\Omega)} \leq \left(\sum\_{\alpha, |\alpha|=1}^P |A\_{\alpha}|^2 \right)^{1/2}
> > > > > > > >                     \left(\sum\_{\alpha, |\alpha|=1}^P\left(\prod\_{i=1}^d
> > > > > > > >                     \left\\|g\_i\right\\|^{\alpha\_i}\_{\mathcal{B}(\Omega)}\right)^2\right)^{1/2} + |A\_0|
> > > > > > > > $$

---

> > > > > > > > > ### Comment · Reviewer_AUeL · 2022-12-11
> > > > > > > > > **Thank you**
> > > > > > > > >
> > > > > > > > > Thanks a lot for the quick reply. So far, I think I am unsure about the suggested form of Theorem 1: Do you also have an updated statement there or does it remain the same?
> > > > > > > > >
> > > > > > > > > Also, just to make sure, I do understand: If we have the case that $L$ is a polynomial, then we can wlog assume $A_0=0$ when considering the PDE $\operatorname{div}(\nabla L(\nabla u)) = 0$. This would imply $C_{\tilde L} = 0$ if I understand it correctly and hence by choosing $u_0=0$ would imply $u^\star = 0$. Am I reading this correctly?

---

> > > > > > > > > > ### Author Response · Authors · 2022-12-11
> > > > > > > > > > **Thank you for your engagement!**
> > > > > > > > > >
> > > > > > > > > > Thank you for the engagement as well!
> > > > > > > > > > The only thing that will change in the main theorem is the bound on the Barron norm of $u_T$. This updated bound is the one we wrote in our reply “Thank you for your response!” that is,
> > > > > > > > > > $$
> > > > > > > > > >     \\|u\_{T}\\|\_{\mathcal{B}(\Omega)} \leq
> > > > > > > > > >                     \left(1 + \frac{\lambda^3}{(C\_p + 1)\Lambda^3} dk^{2}W_0^2 B\_{\tilde{L}}\right)^{p^T+1} \left(\\|u\_0\\|\_{\mathcal{B}(\Omega)}^{p^T} + C\_{\tilde{L}}^{p^{T}}\right)
> > > > > > > > > > $$
> > > > > > > > > > Regarding the clarification, in the case of L being a polynomial your understanding is correct.

---

> > > > > > > > > > > ### Comment · Reviewer_AUeL · 2022-12-12
> > > > > > > > > > > **Reply**
> > > > > > > > > > >
> > > > > > > > > > > Thanks once more for your reply.
> > > > > > > > > > >
> > > > > > > > > > > I now can see that Theorem 1 in the adjusted form might be correct, but I have to wonder about the scope of the setting. The only examples for which you verify Assumption 1 are polynomials, but for polynomial $L$ the target function $u^\star$ becomes trivial. I believe that this should also clearly be stated in your manuscript.
> > > > > > > > > > >
> > > > > > > > > > > Could you give me one non-trivial and relevant example that is covered by your results? I am sure there are some, but without knowing them I can not judge the significance of your results.
> > > > > > > > > > >
> > > > > > > > > > > Note that this also relates to my initial question about non zero right hand sides: solutions of $-\operatorname{div}(\nabla L(\nabla u)) = f$ do not need to be trivial, even for polynomial $L$, e.g., in a special case $-\Delta u = f$. Hence, one possible way to obtain relevant results would be to generalize your analysis to non trivial right hand sides.

---

> > > > > > > > > > > > ### Author Response · Authors · 2022-12-12
> > > > > > > > > > > > **Response!**
> > > > > > > > > > > >
> > > > > > > > > > > > Thank you for your questions and engagement! Encouraged by your enquiries, we looked at our proof and realized that adding $f$ to RHS (which we hope will assuage your concerns) is fairly straightforward.
> > > > > > > > > > > >
> > > > > > > > > > > > Precisely, a solution of a PDE of the form $\text{div}(\nabla L(\nabla u)) = f$ is the minimizer of the variational energy
> > > > > > > > > > > >
> > > > > > > > > > > > $$ E(u) = \int_{\Omega} L(\nabla u(x)) - f(x) u(x) dx$$
> > > > > > > > > > > >
> > > > > > > > > > > > This is easy to check, since the stationary point equation, as in Lemma 1, becomes
> > > > > > > > > > > >
> > > > > > > > > > > > $$dE\[u\](\varphi) = 0, \quad \text{where}\quad dE\left[u\right] (\varphi):= \text{div}(\nabla L(\nabla u)) - f $$
> > > > > > > > > > > >
> > > > > > > > > > > > The statement of Lemma 2 is in fact correct without modification, and the proof is exactly the same one, after one plugs in the above modified expression for $dE$. This also means the proof for convergence (i.e, Lemmas 3, and 4) is exactly the same.
> > > > > > > > > > > >
> > > > > > > > > > > > The only Lemma that changes is Lemma 5, which will now read as follows:
> > > > > > > > > > > >
> > > > > > > > > > > > Lemma 5 (updated) Consider the updates in Equation, if $\tilde{u}\_t \in \Gamma\_{W\_t}$ then for all $\eta \in (0, \frac{\lambda^3}{(C_p+1)\Lambda^3}]$ we have $\tilde{u}\_{t+1} \in \Gamma\_{kW\_t}$ and the Barron norm of $\tilde{u}\_{t+1}$ can be bounded as
> > > > > > > > > > > > $$ \\|\tilde{u}\_{t+1}\\|\_{\mathcal{B}(\Omega)} \leq (1 + \eta d B\_{\tilde{L}})\left(\\|\tilde{u}\_{t}\\|_{\mathcal{B}(\Omega)}^p + C\_{\tilde{L}} + \\|f\\|\_{\mathcal{B}(\Omega)}\right)
> > > > > > > > > > > > $$
> > > > > > > > > > > > Where $\\|f\|\_{\mathcal{B}(\Omega)}$ is the Barron norm of the function $f$.
> > > > > > > > > > > >
> > > > > > > > > > > > This change propagates to Theorem 4 just by modifying the bound of the Barron norm of $u\_T$ as follows:
> > > > > > > > > > > > $$
> > > > > > > > > > > >     \\|u\_{T}\|\_{\mathcal{B}(\Omega)} \leq
> > > > > > > > > > > >                     \left(1 + \frac{\lambda^3}{(C\_p + 1)\Lambda^3} dk^{2}W_0^2 B\_{\tilde{L}}\right)^{p^T+1} \left(\\|u\_0\\|\_{\mathcal{B}(\Omega)}^{p^T} + C\_{\tilde{L}}^{p^{T}} + \\|f\\|\_{\mathcal{B}(\Omega)}^{p^{T}} \right).
> > > > > > > > > > > > $$
> > > > > > > > > > > >
> > > > > > > > > > > > We hope that this clarification helps and would like to sincerely thank the reviewer for the continued engagement and making our submission stronger!!!

---

> > > > > > > > > > > > > ### Comment · Reviewer_AUeL · 2022-12-12
> > > > > > > > > > > > > **Reply**
> > > > > > > > > > > > >
> > > > > > > > > > > > > Dear authors,
> > > > > > > > > > > > >
> > > > > > > > > > > > > thanks once more for your reply, I am glad to see that things can be adjusted, which is no big surprise as the energy is only modified by a linear term. I think your manuscript improved quite a bit, however I am in the difficult situation that a lot of changes had to be made after the end of the rebuttal. Currently for example I still have the following question:
> > > > > > > > > > > > >
> > > > > > > > > > > > > Conside the special case $-\Delta u = f$ with zero boundary values and very small Barron norm $\lVert f\rVert_{\mathcal B}$. Then together with $C_{\tilde L} = 0$ and $u_0=0$, if I understand your updated Theorem correctly, this implies $\lVert u_T\rVert_{\mathcal B}\to0$ for $T\to\infty$ and hence $u_T\to0$ and hence $u^\star=0$. Since the PDE is linear, this would imply $u^\star=0$ for all $f$. Therefore I am wondering, whether I am misinterpreting your results once more.
> > > > > > > > > > > > >
> > > > > > > > > > > > > I am very sorry, for keep asking these questions, but I think a theory paper deserves a proper assessment. Currently I feel that there where too many changes necessary after the end of the rebuttal and too many uncertainties on my end to recommend acceptance right now. Nevertheless, I hope that the manuscript will be published after suitable adjustments.
> > > > > > > > > > > > >
> > > > > > > > > > > > > Best

---

> > > > > > > > > > > > > > ### Author Response · Authors · 2022-12-13
> > > > > > > > > > > > > > **Response!**
> > > > > > > > > > > > > >
> > > > > > > > > > > > > > Hi, thank you for your comment! We definitely appreciate that there have been some changes to the manuscript, thanks to your questions. We would have been happy to make them directly to the draft if editing the manuscript were still allowed. We kept including formal new versions of the lemmas / edits to the proofs in our replies as the next best thing, to be as concrete as possible how these changes would look.
> > > > > > > > > > > > > >
> > > > > > > > > > > > > > Again, we sincerely appreciate all your involvement that has greatly improved our manuscript !
> > > > > > > > > > > > > >
> > > > > > > > > > > > > > Regarding your latest question: you are right, we were trying to respond quickly and had a minor error in unfolding the recursion for the Barron norm bound. The bound should instead be
> > > > > > > > > > > > > >
> > > > > > > > > > > > > > $$
> > > > > > > > > > > > > > \\|\tilde{u}\_{t}\\|\_{\mathcal{B}(\Omega)} \leq (1 + \eta d B\_{\tilde{L}})^{p^T} \\|u\_0\\|\_{\mathcal{B}(\Omega)}^{p^T} \cdot \left( \max\\{1, C\_{\tilde{L}} +  \\|f\\|\_{\mathcal{B}(\Omega)}\\}\right)^{p^T}
> > > > > > > > > > > > > > $$
> > > > > > > > > > > > > >
> > > > > > > > > > > > > > The way this is proven is by writing down the recurrence for the Barron norm:
> > > > > > > > > > > > > > $$
> > > > > > > > > > > > > > \\|\tilde{u}\_{t+1}\\|\_{\mathcal{B}(\Omega)} \leq (1 + \eta d B\_{\tilde{L}})\left(\\|\tilde{u}^p\_{\mathcal{B}(\Omega)} + C\_{\tilde{L}} + \\|f\\|\_{\mathcal{B}(\Omega)}\right)
> > > > > > > > > > > > > > $$
> > > > > > > > > > > > > > Subsequently, we take logarithms on both sides, and after some rearrangement, we get a linear recurrence equation for the log of the Barron norm (which can be easily solved), and finally exponentiating.
> > > > > > > > > > > > > >
> > > > > > > > > > > > > > Again, apologies for the hasty response with a minor issue, and thank you for your continued feedback and interaction!

---

> > > > > > > > > > > > > > > ### Comment · Reviewer_AUeL · 2022-12-13
> > > > > > > > > > > > > > > **Reply**
> > > > > > > > > > > > > > >
> > > > > > > > > > > > > > > Dear authors,
> > > > > > > > > > > > > > >
> > > > > > > > > > > > > > > thanks once more for your very quick reply, I appreciate that you are continuously improving the manuscript.
> > > > > > > > > > > > > > >
> > > > > > > > > > > > > > > In your updated formulation I believe it is again the case that (if $\epsilon_L=0$) setting $u_0=0$ implies $u^\star=0$ for all right hand sides $f\in \mathcal B(\Omega)$, which I think does not hold.
> > > > > > > > > > > > > > >
> > > > > > > > > > > > > > > I know from myself that last minute changes are always risky and prone to errors. This is why I believe the manuscript requires just a little more care to be corrected and updated in order to ensure correctness. I am sorry to say that I still won't raise my score but am sure that once all your updates are carried out nicely the manuscript would be accepted soon. I wish you all the best with this project!

---

### Official Review · Reviewer_23dt · 2022-10-19

**Confidence:** 2
**Correctness:** 3
**Technical Novelty And Significance:** 3
**Empirical Novelty And Significance:** 3
**Recommendation:** 6

**Clarity, Quality, Novelty And Reproducibility:**

The writing is clear, but there is no summary of the limitations.

The methodological innovations are mainly at the theoretical level and hardly appreciated by ICLR readers.


**Strength And Weaknesses:**

Strength

While most previous approaches have studied linear equations, this paper considers a nonlinear family of PDEs and studies nonlinear variational PDEs.

The results go beyond the typical non-parametric bounds on the size of an approximator network that can be easily shown by classical regularity results of the solution to the nonlinear variational PDEs and universal approximation results.

Weaknesses

There is no obvious application for this paper.

This paper considers only the elliptic equations, but the equations that evolve over time are also important.


**Summary Of The Paper:**

This paper studies the representational power of neural networks for approximating solutions to nonlinear PDEs. They focus on a specific class of nonlinear elliptic variational PDEs. They show that a 2-layer neural network can be used to approximate the solution. Thus showing neural networks can evade the curse of dimensionality. This is a theoretical paper; perhaps the application aspect can be strengthened.

**Summary Of The Review:**

This paper presents a theoretical work on neural networks for nonlinear elliptic equations. I am not a mathematics major, and the mathematical proof is difficult for me to understand. It would be more solid to show some applications.

---

> ### Author Response · Authors · 2022-11-10
> **Thank you for your review!**
>
> We thank the reviewer for their comments. Please find answers to your concerns below,
>
> **Re: No obvious applications for this paper:** The key result of our paper gives bounds on the size of a neural network needed to accurately approximate the solution to the PDEs we consider. This has direct *computational* implications, since fitting a neural network is at least as expensive as the neural network required to represent it. It also has *statistical* applications, since many neural network based PDE solvers [E et al, 2018, Lu et al, 2021] are based on Monte Carlo approximations of integrals—the effective sample complexity of which (i.e. number of samples needed for the Monte Carlo approximation to be accurate) depends on the size of the network being fit.
>
> We hope that this alleviates some of the concerns that the reviewer has on the applications of our work.
>
>
> **Re: Only consider elliptic equations, equations that evolve over time are also important:** The main focus of our work is to show results for nonlinear variational PDEs, which can be viewed as describing a “steady state” of a system. In general, time-dependent equations are easier to handle, at least if one is satisfied with a network of size that is linear in T, where T is the amount of time the equation is run for. Such a result can be achieved by discretizing finely enough, and simulating a discrete time step using a neural network. We are not aware of any results that achieve a sublinear in T type of behavior.
>
>
>
> [1] Weinan, E., and Bing Yu. "The deep Ritz method: a deep learning-based numerical algorithm for solving variational problems." Communications in Mathematics and Statistics 6.1 (2018): 1-12.
>
> [2] Lu, Jianfeng, Yulong Lu, and Min Wang. "A priori generalization analysis of the deep ritz method for solving high dimensional elliptic equations." arXiv preprint arXiv:2101.01708 (2021).

---

> > ### Comment · Reviewer_23dt · 2022-11-13
> > **Thanks**
> >
> > More application potential could be discussed in the revised version.

---

> > > ### Author Response · Authors · 2022-11-21
> > > **Thank you!**
> > >
> > > Thank you for your helpful feedback, your engagement, and increasing your score! We re-emphasize the applicability of our results in the revised version of the draft.

---

### Official Review · Reviewer_N9yD · 2022-10-24

**Confidence:** 3
**Correctness:** 3
**Technical Novelty And Significance:** 3
**Empirical Novelty And Significance:** Not applicable
**Recommendation:** 6

**Clarity, Quality, Novelty And Reproducibility:**

The paper is well-written, however, I would suggest a re-organization of the first two sections (see major comment 1) to merge the literature review in the introduction and avoid presenting the main results before the introduction of essential notations. While I am aware of the works on linear PDEs, mentioned by the authors, I have not seen similar attempts on nonlinear PDEs.

**Strength And Weaknesses:**

### Major comments

1. The paper is well-written and nice to read. However, its current organization makes Section 2 not really rigorous (essentially because some notions are introduced later in Section 4). Additionally, the literature review is splitted in two parts between the introduction in Section 1 and related works in Section 3. I suggest a re-organization of Section 1-4 to merge all the literature review into Section 1 by adding:
  - Subsection 1.1 Related works, which includes all the literature review.
  - Subsection 1.2 Contributions, summarizing the essential contributions informally like Theorem 1 but without the notations and definitions introduced later.
2. The authors show the existence of solutions to variational PDEs of the form of (1) with small Barron norm, motivated by a result connecting Barron norm and approximability by a neural network. They also mention in the conclusion that an interesting question is the number of parameters by a neural network like Marwah et al. (2021). However, the main question considered in the introduction (and most interesting due to its connection with the practical works on solving PDEs with neural networks) concern the representability of solutions to PDEs by a small neural network. While the authors show a small complexity of the solution, it would be much more interesting (and impactful) to state a theorem in terms of neural network size after proving Theorem 4.
3. The authors vaguely mention in Section 1 the type of problems modeled by the nonlinear PDEs they consider. It would be good for the clarity of the paper to expand on this by giving precise examples with explicit functions $L$.
4. Would the approach considered by the authors naturally extend to time-dependent PDEs like nonlinear parabolic equations of the form $du/dt - div(grad L(grad u))=0$?
5. The proof structure seems very similar to (Marwhah et al., 2021; Chen et al., 2021), could the authors add some clarifications on the novelty of the proof techniques with respect to these prior works due to the nonlinear class of PDEs considered?

### Minor comments

1. First line of Section 2, $\Omega \subset R^d$, the Poincare constant is not defined. I would write "is such that the Poincare constant Cp satisfies $Cp\geq 1$" and add a footnote defining Cp. [I understand it is defined later in Theorem 2, which is why I suggest a re-organization of the paper].
2. Footnote 2 requires a dot at the end of the sentence.
3. Definition 1, the function space for u is not defined, what is meant by "L is smooth"? In which class of functions does L belong to?
4. Definition 2, the max should be over $1\leq i\leq d$.
5. Theorem 2, add a reference for Poincare inequality?
6. Definition 4 should be "Let F be the set..." or "We define F to be the set...".
7. Definition 5, "Let Gamma be a set...".
8. Theorem 3, would this work for any non-polynomial activation function like the assumption of the universal approximation theorem?
9. Could you comment on the difference with the paper by De Ryck, Jagtap, and Mishra "Error estimates for physics informed neural networks approximating the Navier-Stokes equations", which is analyzing solutions to the Navier-Stokes equations?

**Summary Of The Paper:**

In this paper, the authors consider the potential approximability of solutions to PDEs by neural networks. This topic has attracted significant interest in the past few years and a key question, as noted by the authors, is to understand which class of PDEs give rise to solutions that can be efficiently represented by a neural network without the curse of dimensionality, i.e. when the network size doesn't grow exponentially with the spatial dimension. The authors expand upon the previous works on linear PDEs by consider nonlinear variational PDEs of the form $-\nabla\cdot(\nabla L(\nabla u))=0$ and show that the solution has low complexity when $L$ does. Here, the complexity is formulated in terms of the Barron norm of the solution, which is a standard tool in the deep learning and PDE literature.

**Summary Of The Review:**

The authors generalize existing results on the approximation of solutions to elliptic PDEs by neural networks to nonlinear variational PDEs. The paper is well-written and a good addition to the literature on this topic but would benefit from a re-organization of the structure and clarification of the novelty of the proofs compared to prior works. In addition, the authors show that solutions to the nonlinear PDEs have low-complexity but do not continue to derive a result on the size of the neural networks needed to approximate them. I would recommend the paper for publication after a revision addressing the major and minor comments listed above.

---

> ### Author Response · Authors · 2022-11-11
> **Thank you for your review!**
>
> We thank the reviewer for their detailed comments and feedback. We have added a statement that combines the result from Theorems 3 and 4 to give the final bound on the neural network size in Remark 3 (the draft edits are shown in blue).
>
> **Re: Extension to time-dependent PDE like nonlinear parabolic PDEs**
> The main focus of our work is to show results for nonlinear variational PDEs, which can be viewed as describing a “steady state” of a system. In general, time-dependent equations are easier to handle, at least if one is satisfied with a network of size linear in T, where T is the amount of time the equation is run for. Such a result can be achieved by discretizing finely enough, and approximating a discrete time step using a neural network. We are not aware of any results that achieve a sublinear in T type of behavior.
>
>
> **Re: Proof structure very similar to Marwah et al (2021), Chen et al (2021)**
> While the proof idea for gradient descent in an appropriate Hilbert space follows from previous work by Marwah et al (2021) and Chen et al (2021), a significant amount of new technical machinery is required to make the leap to nonlinear elliptic variational PDEs. For example, teasing out the right proxies of convexity of the variational loss (Lemma 2), as well as choosing the right preconditioner to get progress in the “right norm” (see Lemma 3, and the comments before it) are completely new contributions.
>
>
> **Re: Difference with the paper with De Ryck, Jagtap and Mishra**
> Thank you for bringing this work to our attention! This paper also studies the number of parameters required by a neural network to approximate a nonlinear PDE (in this case, Navier-Stokes). However, the setup considered in this paper is that of physics-informed neural networks (PINNS) where the physical constraints are added as soft-loss to be minimized by a neural network. The loss function considered in our paper is more in the flavor of variational losses used in methods like Deep Ritz/Galerkin methods.
> Furthermore, the analysis performed in their work results in a neural network with number of parameters that generally scales exponentially in the input dimension d, i.e., the final rate is  O(c^d) for some constant c—therefore does not evade the curse of dimensionality.
>
> **Re: Theorem 3, would this work for any non-polynomial activation function like the assumption of the universal approximation theorem?**
> Yes, the result should work for any polynomial activation, as long as it is sigmoidal.
>
>
> **Re: Exposition and Organization.**
> Thank you for your suggestions on the exposition and organization of the paper. We have implemented many of your suggestions on language and fixed the typos that you point out. Concerning the broader organization of the paper, our aim is to convey the main message and the result within the introduction (Section 1) itself—without getting overwhelmed by relatively involved definitions and notations. This is an attempt to make sure that while going through the related work section (Section 2) the reader has an understanding of our main result and proof sketch, so that they can place the contributions when juxtaposed with prior work.

---

> > ### Comment · Reviewer_N9yD · 2022-11-17
> > **Response to authors' comment**
> >
> > I thank the authors for the response which answers most of my comments. I would suggest to add a paragraph in the final version expanding upon point 4 of my original review to give concrete examples of applications to motivate this theoretical work for a broaden audience.

---

> > > ### Author Response · Authors · 2022-11-21
> > > **Thank you for your response**
> > >
> > > Thank you for your feedback and engagement! We will add more examples for specific choices of the function L in the final version of our draft.

---

### Official Review · Reviewer_4yVB · 2022-10-26

**Confidence:** 4
**Correctness:** 4
**Technical Novelty And Significance:** 3
**Empirical Novelty And Significance:** Not applicable
**Recommendation:** 8

**Clarity, Quality, Novelty And Reproducibility:**

The paper is very clear, at least for someone who is familiar with this sub-area.  The main argument is in the supplementary materials rather than the main body, and I have not checked all the details.  Nonetheless, this seems like a fairly standard type of argument for this flavor of results, and I believe that it is probably correct, and a nice result.  Extending these types of analyses from linear to nonlinear PDEs is generally a useful thing to do, and I am not aware of other results of this form for this case.  The paper is purely theoretical, so there are no issues of reproducibility here.

**Strength And Weaknesses:**

The paper is a nice result, clearly written.  The proof tools are standard for this type of PDE analysis, but they are well used here.

The result does not tell us anything about deeper neural networks, nor does it seem to tell us how to find the optimal approximating networks that satisfy the bound -- we just know that such networks must exist.

I would like to see this work out in the literature somewhere, and it is certainly relevant to people trying to build neural network schemes for the solution of PDEs arising in science.  But I admit that if I was looking for a result of this form, I would probably usually start in a journal on approximation theory or PDE analysis rather than in ICLR or similar conference proceedings.

**Summary Of The Paper:**

The paper is fundamentally a PDE analysis result involving the regularity of solutions to (nonlinear) variational problems with respect to the Barron norm when the function involved in the variational problem (to within an epsilon approximation in the H^1 seminorm) satisfies a Holder-like condition (with respect to the Barron norm).  The proof is via analysis of an iterative approximation scheme (preconditioned gradient descent).  This is relevant for variational approximation schemes using neural networks as a trial space, because bounds on the Barron norm of the solution correspond to bounds on the size of a two-layer neural network needed to represent the solution to a given accuracy.

**Summary Of The Review:**

This is a nice result, and I would like to see it published somewhere, though it would not be obvious to me that an ML conference is the venue where it would have the best reception.  Complexity of approximation is important for understanding numerical schemes for solving PDEs generally, and there are relatively few such results around approximation schemes that involve neural networks.

---

> ### Author Response · Authors · 2022-11-10
> **Thank you for your review!**
>
> We thank the reviewer for your very encouraging review and feedback! We are glad that you found our results nice, and the writing clear. We fully agree with you that providing results on the training dynamics of fitting the neural networks, as well as more fine-grained architecture design are interesting directions for future work.
>
> **Regarding ICLR being an appropriate venue for the work:** We note that the two most directly related works to ours, Marwah et al. and Chen et al., were both published in a flagship machine learning conference (NeurIPS 2021). Given the recent interest of the machine learning community in applying neural networks to PDE solvers, we think work like ours is well-placed in a machine learning venue.

---

### Decision · Program_Chairs · 2023-01-20

**Decision:**

Reject

**Justification For Why Not Higher Score:**

The authors presented a theory for which there are no non-trivial examples satisfying their assumptions.

**Justification For Why Not Lower Score:**

N/A

**Metareview: Summary, Strengths And Weaknesses:**

to PDEs by neural networks. This topic has attracted significant interest in the past few years and a key question, as noted by the authors, is to understand which class of PDEs give rise to solutions that can be efficiently represented by a neural network without the curse of dimensionality, i.e. when the network size doesn't grow exponentially with the spatial dimension. The authors expand upon the previous works on linear PDEs by consider nonlinear variational PDEs of the form $-\nabla \cdot ( \nabla L ( \nabla u)) = 0$  and show that the solution has low complexity when $L$ does. Here, the complexity is formulated in terms of the Barron norm of the solution, which is a standard tool in the deep learning and PDE literature.

All reviewers thought this paper make an interesting contribution on the topic of approximability of solutions to PDEs by neural networks, which was previously mostly limited to linear PDEs. While reviewer 4yVB questioned the choice of venue for this kind of work, the authors have provided a satisfying answer to this point in their rebuttal. Yet, despite three positive reviewers, a detailed technical discussion between the authors and reviewer AUeL highlighted new technical issues with the statements in the revised manuscript. These appear to be fixable according to the discussion, but there was a remaining important question from reviewer 4yVB for which they did not get any convincing answer yet (see more details below). The AC agrees that this concern needs to be appropriately addressed in a revision before the paper is acceptable.

== More details on the concern from 4yVB (see discussion for details)

The theorems in the paper crucially depend on Assumption 1; the only example of functions provided by the authors to satisfy Assumption 1 are polynomials; but yet 4yVB pointed out that $u^*$ becomes trivial in this case (i.e. the zero solution), thus providing a vacuous theory. The authors should provide at least one example covered by the theory for which the approximated function is non-trivial.